# A common human *MLKL* polymorphism confers resistance to negative regulation by phosphorylation

Sarah E. Garnish[1,2], Katherine R. Martin [1,2], Maria Kauppi [1,2], Victoria E. Jackson [1,2], Rebecca Ambrose[3,4], Vik Ven Eng[3,5], Shene Chiou[1,2], Yanxiang Meng [1,2], Daniel Frank[1], Emma C. Tovey Crutchfield [1,6], Komal M. Patel[1], Annette V. Jacobsen [1,2], Georgia K. Atkin-Smith[1,2], Ladina Di Rago[1,2], Marcel Doerflinger [1,2], Christopher R. Horne [1,2], Cathrine Hall[1], Samuel N. Young[1], Matthew Cook [7,12], Vicki Athanasopoulos [8], Carola G. Vinuesa[7,8,13,14,15], Kate E. Lawlor [3,4], Ian P. Wicks [1,2], Gregor Ebert [9], Ashley P. Ng [1,2,10], Charlotte A. Slade[1,2,11], Jaclyn S. Pearson [3,4,5], André L. Samson [1,2], John Silke [1,2], James M. Murphy [1,2] & Joanne M. Hildebrand [1,2] ✉

Across the globe, 2-3% of humans carry the *p.Ser132Pro* single nucleotide polymorphism in *MLKL*, the terminal effector protein of the inflammatory form of programmed cell death, necroptosis. Here we show that this substitution confers a gain in necroptotic function in human cells, with more rapid accumulation of activated MLKL^S132P in biological membranes and MLKL^S132P overriding pharmacological and endogenous inhibition of MLKL. In mouse cells, the equivalent *Mlkl S131P* mutation confers a gene dosage dependent reduction in sensitivity to TNF-induced necroptosis in both hematopoietic and non-hematopoietic cells, but enhanced sensitivity to IFN-β induced death in non-hematopoietic cells. In vivo, *Mlkl^S131P* homozygosity reduces the capacity to clear *Salmonella* from major organs and retards recovery of hematopoietic stem cells. Thus, by dysregulating necroptosis, the S131P substitution impairs the return to homeostasis after systemic challenge. Present day carriers of the *MLKL S132P* polymorphism may be the key to understanding how MLKL and necroptosis modulate the progression of complex polygenic human disease.

Necroptosis is a caspase independent form of programmed cell death that originated as a defense against pathogens[1–5]. Highly inflammatory in nature, necroptosis results in the permeabilization of biological membranes and the release of cytokines, nucleic acids, and intracellular proteins into the extracellular space[6]. Necroptosis is induced by danger- or pathogen-associated molecular patterns that signal via transmembrane receptors or intracellular pattern recognition receptors[7–11]. Of the various initiating stimuli, the most well studied necroptotic pathway is downstream of tumor necrosis factor receptor

1 (TNFR1)[12]. In physiological contexts that favor low cellular inhibitor of apoptosis protein 1 (cIAP1) and caspase-8 activity, TNFR1 signals culminate in the formation of a high molecular weight platform called the necrosome that is nucleated by heterooligomeric RIPK1 and RIPK3[13–15]. Here, the terminal executioner protein, MLKL, is phosphorylated and activated by its upstream kinase, RIPK3[16–18]. Following phosphorylation, MLKL dissociates from RIPK3, oligomerizes, and is trafficked to biological membranes where it interacts with Phosphatidylinositol Phosphates[19–28]. In human cells, association of activated MLKL

oligomers with biological membranes can be inhibited by the synthetic compound necrosulfonamide[16,22,28] or inhibitory phosphorylation of MLKL at Serine 83[29]. Pharmacological or mutation driven disruption at any major necroptotic signaling checkpoint compromises a cell's capacity to execute necroptosis.

In mouse studies, MLKL-mediated cell death has been implicated as a driver or suppressor of diseases spanning almost all physiological systems depending on the pathological context. The generation of *Mlkl* gene knockout (*Mlkl*[−/−]), knock-in and conditional knockout mouse models have enabled the role of necroptosis in infectious and non-infectious challenges to be dissected in physiological detail[30]. Interestingly, genetic deletion of *Mlkl* has no overt developmental or homeostatic effects, with the exception of a reduction in age-related sterile inflammation in female mice[18,31,32]. This is in direct contrast with two mouse models harboring *Mlkl* point mutations that dysregulate MLKL activation, *Mlkl*[D139V] and *Mlkl*[S83G], which exhibit early neonatal death and severe inflammatory phenotypes[29,33]. Altogether, these observations suggest that while constitutive absence of MLKL-mediated death is benign, imbalanced execution of necroptotic cell death is deleterious.

Consistent with this notion, more than 20 unique disease-associated human germline gene variants in the core necroptotic machinery, encompassing *RIPK1*, *RIPK3*, *MLKL*, have been identified[34,35]. In one family, a haplotype including a rare *MLKL* loss-of-function gene variant (*p.Asp369GlufsTer22*, rs561839347) is associated with a severe and progressive novel neurogenerative spectrum disorder characterized by global brain atrophy[36]. A more frequent *MLKL* loss-of-function gene variant (*p.Gln48Ter*, rs763812068) was found to be >20 fold enriched in a cohort of Hong Kong Chinese patients suffering from Alzheimer's disease[37] and common variants that cluster around the MLKL brace region were shown to be enriched in trans in a cohort of Chronic Recurrent Multifocal Osteomyelitis patients[33]. More recently, a hypomorphic *MLKL* missense gene variant (*p.G316D*, rs375490660) was reported to be associated with Maturity Onset Diabetes of the Young[38].

Here we present the cellular and physiological characterization of a serine to proline missense polymorphism at MLKL amino acid 132 (*p.Ser132Pro*; S132P). The S132P polymorphism is the third most frequent human *MLKL* missense coding variant in the gnomAD database, a large repository of whole genome and exome sequence data from humans of diverse ancestry[39]. To examine the potential human disease-causing effects of this *MLKL* variant, we exogenously expressed *MLKL*[S132P] in human cell lines and introduced the mouse counterpart variant (*Mlkl*[S131P]) into a genetically modified mouse model, revealing that this polymorphism confers MLKL gain-of-function in a cell- and stimulus-dependent manner. This MLKL gain-of-function manifests in in vivo changes to the immune response, impaired bacterial clearance, and defective emergency hematopoiesis. These observed phenotypes provide important insights into how this highly frequent human *MLKL* S132P polymorphism may contribute to the progression of complex disease.

## Results

### Carriers of the *p.Ser132Pro* polymorphism exhibit diverse inflammatory disease profiles

With a global minor allele frequency (MAF) of 0.0138, the *MLKL S132P* (rs35589326) polymorphism is predicted to be carried by 2–3% of the human population. It has not been detected in individuals assigned East Asian ancestry, is rare in individuals of African or Latino/Admixed American ancestry, and is carried by an estimated 6–7% of individuals of Ashkenazi Jewish ancestry (MAF 0.0315) (www.gnomAD.com, February 2023) (Fig. 1a). Notably, Ser132 is highly conserved across species (Fig. 1b).

Two heterozygous carriers of the *MLKL S132P* polymorphism were identified in an Australian registry of patients suffering from immune

related disease. Patient 1, a female of South American heritage, was diagnosed with SAPHO (synovitis, acne, pustulosis, hyperostosis, osteitis) syndrome (inheritance chart unavailable). Patient 2, a female of European heritage, was diagnosed with systemic IgG4 disease (Supplementary Fig. 1A). Both patients have one or more immediate family members carrying the *MLKL S132P* polymorphism who were also diagnosed with inflammatory diseases in early adulthood. Following whole genome sequencing, Patient 1 was found to exhibit a region of loss of heterozygosity (5:96031569–5:96364063) which covers *CAST*, *ERAP1*, *ERAP2* and *LNPEP*. These genes have been previously associated with inflammatory disease[40–44]. Patient 2 does not carry any other predicted pathogenic gene variants with a MAF < 0.0005 (IUIS human Inborn Errors of Immunity). Primary peripheral blood mononuclear cells (PBMC) isolated from patient 2 showed reduced MLKL protein levels, accompanied by a nominal increase in pro-inflammatory cytokine production in response to the Toll-like receptor (TLR) agonists, lipopolysaccharide and poly I:C, relative to age and sex matched healthy donor PBMCs not carrying the *MLKL S132P* polymorphism (Supplementary Fig. 1B, C).

### *MLKL*[S132P] confers resistance to chemical and natural regulatory inhibition

To examine any changes to MLKL function conferred by this common polymorphism, we stably transduced parental wild-type (WT) and *MLKL*[−/−] versions of the human colonic epithelial cell line HT29 with doxycycline inducible *MLKL*[WT] and *MLKL*[S132P] gene expression constructs. We also included the T357/S358 phosphosite mutant *MLKL*[TSEE], previously shown to be inactive[20,45], as a negative control for necroptosis induction. Cells expressing *MLKL*[S132P] died with similar kinetics to *MLKL*[WT] following necroptotic stimulation (TNF, Smac mimetic compound A and pan-caspase inhibitor IDN-665; TSI) and, as expected, *MLKL*[TSEE] reconstituted cells were resistant to necroptotic cell death (Fig. 1c, Supplementary Fig. 1D, E). Interestingly, in the presence of the MLKL inhibitor necrosulfonamide (NSA), TSI-stimulated cells expressing *MLKL*[S132P] exhibited higher levels of cell death than their *MLKL*[WT] counterparts at later timepoints (Fig. 1c, Supplementary Fig. 1D). We termed this override of MLKL inhibition 'breakthrough' cell death. Breakthrough death also occurred when *MLKL*[S132P] was co-expressed with endogenous *MLKL* and increased when *MLKL*[S132P] expression was augmented by higher doses of doxycycline (Fig. 1c, Supplementary Fig. 1D, E). We also observed breakthrough death in both wild-type and *MLKL*[−/−] forms of the human monocytic cell line U937, indicating this is not a cell type specific phenomenon (Supplementary Fig. 1F, G). Breakthrough death was neither due to differences in MLKL expression, nor changes in MLKL phosphorylation (T357/S358), because both were equivalent between *MLKL*[WT] and *MLKL*[S132P] reconstituted cells (Fig. 1d, Supplementary Fig. 1H, I). Instead, cellular fractionation experiments suggest that breakthrough death was likely due to enhanced association of MLKL[S132P] with biological membranes (Fig. 1e, Supplementary Fig. 1J).

The small molecule inhibitor, NSA, blocks human MLKL activity through covalent modification of Cysteine 86, located in the MLKL four helix-bundle, executioner domain (Fig. 1f)[16,28]. Recently, it was discovered that MLKL can be endogenously phosphorylated at the proximal residue, Serine 83, and this phosphorylation event plays a species conserved inhibitory role in both human (S83) and mouse (S82) (Fig. 1f)[29]. Like NSA, phosphorylation of MLKL S83 does not alter the capacity of RIPK3 to phosphorylate MLKL at S357/T358 but prevents necroptosis by blocking the association of MLKL with cellular membranes[29]. To test whether phosphorylation at S83 is, like NSA, less effective at inhibiting MLKL[S132P] induced death, we created stable HT29 (*MLKL*[−/−] and wild-type) cell lines that exogenously express gene constructs encoding *MLKL*[WT], *MLKL*[S132P], *MLKL*[S83D] (phosphomimetic) and double mutant *MLKL*[S83D,S132P]. Consistent with recent studies[29], the phosphomimetic mutation of S83, *MLKL*[S83D], ablated the capacity of

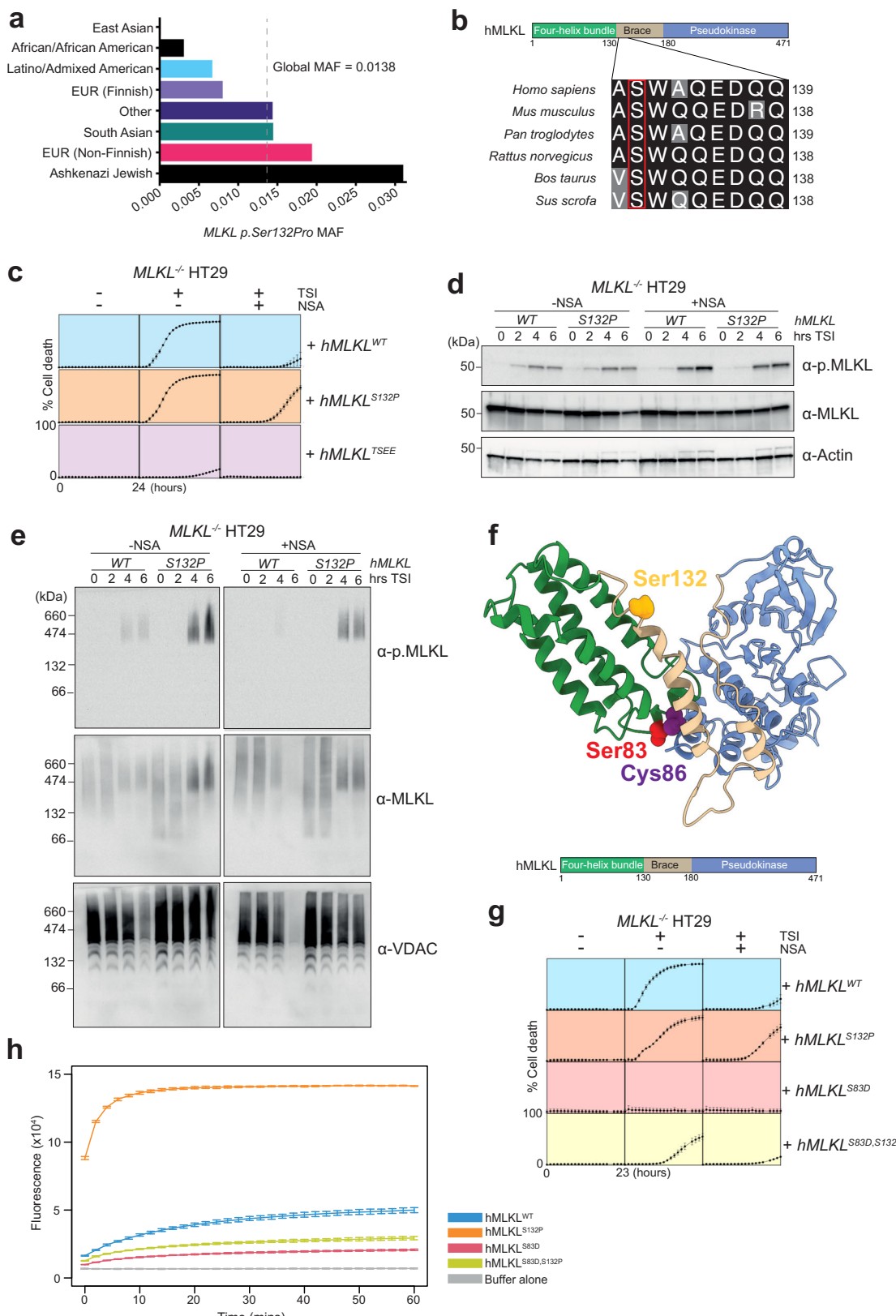

MLKL to execute necroptotic cell death (Fig. 1g). Also, as previously reported, the inhibitory effect of *MLKL^S83D* overrode the activating effects of endogenous MLKL phosphorylation at Ser357/Thr358 (Fig. 1g, Supplementary Fig. 1K, L). Exogenous expression of *MLKL^S83D* also reduced total levels of necroptotic cell death in the presence of endogenous MLKL, indicating that this S83 phosphomimetic acts in a

dominant negative manner (Supplementary Fig. 1L). Strikingly, combining the MLKL^S132P and MLKL^S83D substitutions to create the compound mutant MLKL^S83D, S132P restored necroptotic cell death responsiveness, albeit with reduced kinetics and reduced maximal cell death when compared to MLKL^S132P alone (Fig. 1g). The *MLKL^S132P* variant also partially overcomes the dominant negative effect of S83

**Fig. 1 | MLKL$^{S132P}$ executes cell death in the presence of necrosulfonamide or Ser83 inhibition. a** Minor allele frequency (MAF) of *MLKL p.Ser132Pro* according to the gnomAD database, stratified by ancestry. **b** Sequence alignment of conserved serine in the MLKL orthologs across different species. **c** *MLKL$^{WT}$*, *MLKL$^{S132P}$* and *MLKL$^{TSEE}$* expression was induced in *MLKL$^{-/-}$* HT29 cells with 100 ng/ml doxycycline (Dox) and cells treated with necroptotic stimulus (TNF, Smac mimetic, IDN-6556; TSI) in the presence or absence of MLKL inhibitor Necrosulfonamide (NSA; 1 μM). Percentage SYTOX Green positive (dead) cells quantified using IncuCyte SX5 live cell imager. Cell lines were assayed in *n* = 3 independent experiments, with error bars indicating the mean ± SEM. **d** Western blot analyses of whole cell lysates and **e** Blue-Native PAGE analyses of crude membrane fractions taken post TSI stimulation in the presence or absence of NSA from *MLKL$^{-/-}$* HT29 cells expressing *MLKL$^{WT}$* or *MLKL$^{S132P}$*. **f** S83 (red), C86 (purple) and S132 (gold) highlighted as spheres on cartoon representation of human MLKL (four-helix bundle domain (green), brace helices (beige), and pseudokinase domain (blue)). Homology model generated from PDB:2MSV [https://doi.org/10.1016/j.str.2014.07.014] and PDB: 4MWI [https://doi.org/10.1042/BJ20131270] of human MLKL, which were aligned using the full-length murine crystal structure (PDB:4BTF [https://doi.org/10.1016/j.immuni.2013.06.018])[18,71,72]. **g** Human *MLKL* expression was induced in *MLKL$^{-/-}$* HT29 cells with doxycycline (100 ng/ml) and treated TSI in the presence or absence of NSA (1 μM). Percentage SYTOX Green positive (dead) cells quantified using IncuCyte SX5 live cell imager. Cell lines were assayed in *n* = 4 independent experiments, with error bars indicating the mean ± SEM. **h** Liposome dye release assays using 0.5 μM recombinant full-length MLKL$^{WT}$, MLKL$^{S132P}$, MLKL$^{S83D}$, and MLKL$^{S83D,S132P}$. Release of 5(6)-Carboxyfluorescein was measured by fluorescence (485 nm excitation wavelength, 535 nm emission wavlength) every 2 min over 60 min. Data represent mean ± SD of triplicate measurements, representative of three independent assays. **d**, **e** Blot images are representative of at least three independent experiments. Source data are provided as a Source Data file.

phosphomimetic mutation in cells that endogenously express wild-type *MLKL* (Supplementary Fig. 1L).

To further dissect how the MLKL$^{S132P}$ substitution facilitates gain-of-function, we used liposome dye release assays to test the membrane-damaging capacity of recombinant full-length MLKL$^{WT}$, MLKL$^{S132P}$, MLKL$^{S83D}$ and MLKL$^{S83D, S132P}$ expressed and purified from insect cells. We found that the membrane-damaging capacity of recombinant MLKL$^{S132P}$ was increased, and MLKL$^{S83D}$ reduced, relative to MLKL$^{WT}$ (Fig. 1h). Consistent with our observations in cells, the combination mutant MLKL$^{S83D, S132P}$ displayed a membrane-damaging capacity greater than MLKL$^{S83D}$ alone but reduced in comparison to MLKL$^{WT}$ (Fig. 1h). Our results suggest S132P promotes membrane association, and this is likely to contribute to the gain-of-function we observed in vitro under pharmacological and natural inhibition.

Given the clear gain-of-function conferred to MLKL in the context of simulated inhibitory phosphorylation (MLKL$^{S83D, S132P}$), we questioned whether mutations that resemble this phosphomimetic mutant may occur naturally. According to the gnomAD database (www.gnomAD.com, Jan 2023), no individuals have been recorded with a polymorphism that encodes the *p.Ser83Asp* (S83D) replacement. However, there are individuals that carry closely related changes, *p.Ser83Cys* (MAF 3.54e10$^{-5}$) and *p.Arg82Ser* (MAF 1.57e10$^{-5}$). We created stable *MLKL$^{-/-}$* HT29 cell lines that exogenously express gene constructs encoding *MLKL$^{S83C}$* and *MLKL$^{R82S}$* and assessed their capacity to execute necroptosis. Cells expressing *MLKL$^{S83C}$* died with similar kinetics to *MLKL$^{WT}$* following necroptotic stimulation however, *MLKL$^{R82S}$* reconstituted cells were resistant to cell death (Supplementary Fig. 1M). Consistent with our observations for *MLKL$^{S83D}$*, *MLKL$^{R82S}$* was phosphorylated at Ser357/Thr358, and the *MLKL$^{R82S,S132P}$* compound mutant restored necroptotic killing function (Supplementary Fig. 1M, N). Whilst gene variants at or adjacent to the S83 inhibitory phosphorylation site are less frequent than the *MLKL S132P* polymorphism in humans, they nonetheless strengthen the precedent for genetically encoded diversity in human MLKL function, and the potential for functionally synergistic or neutralizing combinations of *MLKL* gene variants.

### *MLKL$^{S131P}$* mice exhibit differences in steady state immune cell populations

To address if the *MLKL S132P* polymorphism contributes to immunoinflammatory disease, we performed detailed histological, immunophenotypic, and experimental analyses of a CRISPR generated knockin mouse which carried the orthologous mutation, *Mlkl$^{S131P}$*. *Mlkl$^{S131P}$* heterozygotes and homozygotes were born according to the expected Mendelian ratios and had normal lifespans (Fig. 2a, b). The healthy presentation of *Mlkl$^{S131P}$* mice contrasts with the lethal phenotypes of mice engineered to encode *Mlkl$^{D139V}$* and *Mlkl$^{S83G}$* mutations[29,33]. Wild-type, *Mlkl$^{S131P}$* heterozygote and homozygote mice had comparable body weight and no gross histological differences up to 9 months of age (Fig. 2c, Supplementary Fig. 2A). Notably, distinct from *Mlkl$^{D139V}$* or *Mlkl$^{S83G}$* homozygotes, no inflammation was detected in the salivary glands, mediastinum, liver or lungs (Supplementary Fig. 2A)[29,33]. No differences in the number of circulating platelets, red blood cells, or white blood cells were observed between wild-type, *Mlkl$^{S131P}$* heterozygote and homozygote mice across age (Supplementary Fig. 2B–D). However, reduced numbers of classical Ly6C$^{hi}$ 'inflammatory' monocytes in the bone marrow, but not in the secondary lymphoid organs (spleen and inguinal lymph nodes), were observed in *Mlkl$^{S131P}$* homozygotes and heterozygotes relative to wild-type littermate controls (Fig. 2d–i, Supplementary Fig. 2E, J, O). In the secondary lymphoid organs, *Mlkl$^{S131P}$* homozygotes had a small but significant increase in splenic CD4$^+$ T cells and B cells relative to wild-type littermate controls (Fig. 2g, Supplementary Fig. 2M, N). Compared to wild-type controls, *Mlkl$^{S131P}$* heterozygotes had significant increases in splenic and lymph node B cell populations, as well as splenic CD8$^+$ T cells and Ly6C$^{lo}$ monocytes (Fig. 2g–i, Supplementary Fig. 2J–S). All other innate and adaptive immune cell populations were comparable between genotypes in the bone marrow, spleen, and inguinal lymph nodes (Fig. 2d–i, Supplementary Fig. 2E–S). Further, wild-type, heterozygote and homozygote mice had comparable levels of plasma cytokines at steady state (Fig. 2j).

To investigate the underlying cause of reduced Ly6C$^{hi}$ monocytes in the bone marrow of *Mlkl$^{S131P}$* homozygote mice, we examined both the abundance of different myeloid stem cell populations, and their capacity to differentiate and form colonies ex vivo. *Mlkl$^{WT/WT}$* and *Mlkl$^{S131P/S131P}$* hematopoietic stem cells exhibited equivalent capacity to generate blast, eosinophil, granulocyte, granulocyte-macrophage, and megakaryocyte colonies under all stimulations investigated (Fig. 2k). While the number and composition of myeloid progenitor cell populations in the bone marrow were not significantly different (Supplementary Fig. 2t), compared to *Mlkl$^{WT/WT}$*, *Mlkl$^{S131P/S131P}$* bone marrow gave rise to an increased number of macrophage colonies under SCF, IL-3 and EPO combined stimulation (Fig. 2k).

We next sought to address whether mouse MLKL$^{S131P}$ exhibited a gain-in-function comparable to our finding for MLKL$^{S132P}$ in human cells. Primary and immortalized fibroblasts isolated from the dermis of *Mlkl$^{S131P}$* homozygote, heterozygote and wild-type mice were examined for their relative capacity to die in the presence or absence of death stimuli (Fig. 2l, Supplementary Fig. 2U, V). We did not observe any differences in TNF-induced apoptosis (TNF and Smac mimetic; TS) between *Mlkl$^{S131P/S131P}$* and *Mlkl$^{WT/WT}$* immortalized MDFs (Fig. 2l, Supplementary Fig. 2V). However, *Mlkl$^{S131P/S131P}$* MDFs showed a diminished capacity to undergo TSI-induced (TNF, Smac mimetic compound A, pan-caspase inhibitor IDN-6556; TSI) necroptotic cell death (Fig. 2l, Supplementary Fig. 2V). This was driven by reduced MLKL protein levels in MDF cells with the *Mlkl$^{S131P}$* allele (Fig. 2m, Supplementary Fig. 2W). Immortalized MDFs, and to a lesser extent primary MDFs, did show clear *Mlkl$^{S131P}$* allele-dependent sensitivity to IFN-β, a strong

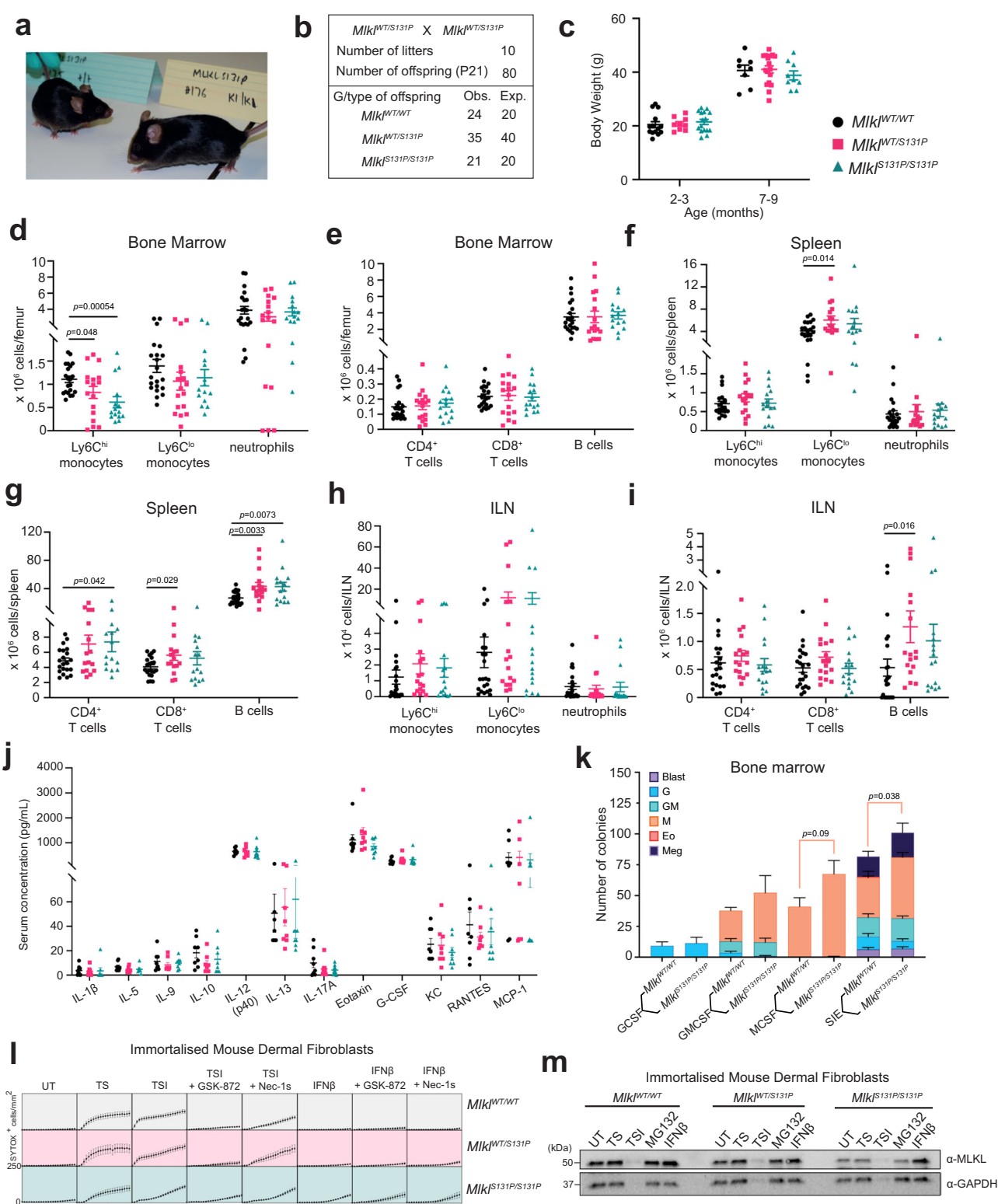

inducer of *Mlkl* gene expression in mice (Fig. 2l, Supplementary Fig. 2U, V)[46]. This sensitivity was further enhanced by TNF and was refractory to inhibitors of RIPK1 (Nec-1s) or RIPK3 (GSK'872) (Fig. 2l, Supplementary Fig. 2U, V). Furthermore, Mlkl[S131P] forms high molecular weight oligomers following combined stimulation with IFNβ and TNF that are present in the crude membrane fraction of cells (Supplementary Fig. 2X). These results are analogous to cells encoding the constitutively-active *Mlkl[D139V]* mutant and cells exogenously expressing inducible *Mlkl[S131P]*[33]. This supports the notion that MLKL[S131P] has

constitutive RIPK3-independent activity when endogenously expressed in mouse cells[33]. These endogenous *Mlkl[S131P]* observations were not limited to fibroblasts. In bone marrow derived macrophages (BMDM), *Mlkl[S131P/S131P]* cells also exhibited diminished sensitivity to TSI-induced necroptosis and endogenously produced MLKL[S131P] was present at reduced levels in comparison to MLKL[WT] (Supplementary Fig. 2Y, Z). In contrast to immortalized dermal fibroblasts, BMDMs derived from *Mlkl[S131P]* homozygotes did not undergo cell death in the presence of IFN-β alone, despite clear IFN-β induced upregulation of MLKL protein

**Fig. 2 | In mice, *Mlkl^{S131P}* homozygosity is tolerated but results in steady state immune cell population differences. a** Macroscopic appearance of *Mlkl^{WT/WT}* and *Mlkl^{S131P/S131P}* mice at 8–12 weeks of age. **b** *Mlkl^{S131P/S131P}* mice are born according to Mendelian ratios as observed in the distribution of genotypes from *Mlkl^{WT/S131P}* heterozygous intercrosses. **c** Body weight of *Mlkl^{WT/WT}*, *Mlkl^{WT/S131P}*, and *Mlkl^{S131P/S131P}* mice at 3–4 or 7–9 months of age. Each dot represents an individual mouse (*n* = 14, 9, 16, 8, 17, 9, mean ± SEM). **d–i** Flow cytometry quantification of innate (Ly6C^{hi} monocytes, Ly6C^{lo} monocytes and neutrophils) and adaptive (CD4^+ T cells, CD8^+ T cells and B cells) in the (**d, e**) bone marrow (*n* = 21, 17, 15), (**f, g**) spleen (*n* = 21, 16, 15), and (**h, i**) inguinal lymph nodes (*n* = 22, 17, 16) of 8–12-week-old *Mlkl^{WT/WT}*, *Mlkl^{WT/S131P}*, and *Mlkl^{S131P/S131P}* mice. Each symbol represents one biologically independent mouse and error bars represent mean ± SEM. **j** Multiplex measurement of plasma cytokines from 6–12 week old mice. Each symbol represents one individual mouse sampled, with mean ± SEM of *n* = 8. **k** Type and number of colonies from 25,000 unfractionated bone marrow cells cultured in G-CSF (10³ U/mL), GM-CSF (10³ U/mL), SIE [SCF (100 ng/mL), IL-3 (10 ng/mL), EPO (2 U/mL)] were scored after 7 days. Error bars represent mean ± SEM for *n* = 4. **l, m** Immortalized mouse dermal fibroblasts (MDF) were isolated from *Mlkl^{WT/WT}*, *Mlkl^{WT/S131P}*, and *Mlkl^{S131P/S131P}* mice and stimulated as indicated for 22 hours for quantification of SYTOX-positive cells using IncuCyte SX5 live cell imager (**l**) or for 6 hours for western blot analysis (**m**). Death data represent mean ± SEM, *n* = 4 *Mlkl^{WT/WT}*, 4 *Mlkl^{WT/S131P}*, 5 *Mlkl^{S131P/S131P}* cell lines examined over 1, 2, or 3 independent experiments. *P* value calculated using an unpaired, two-tailed Students t test. Source data are provided as a Source Data file.

relative to untreated cells (Supplementary Fig. 2Y, Z). Overall, these data show that, across different cell types, MLKL^{S131P} is not toxic at steady state, but upon different stimuli, both gain- and loss-of-sensitivity to necroptotic cell death is evident. This suggests that any functional deficits or enhancements in carriers of the *MLKL S132P* polymorphism are likely to be diverse, with cell- and/or context-specific manifestations.

## Emergency hematopoiesis is defective in *Mlkl^{S131P/S131P}* mice

Differences in steady state immune cell populations suggest that overt phenotypes may develop in *Mlkl^{S131P}* homozygotes following experimental challenge. Specifically, reduced numbers of steady state Ly6C^{hi} monocytes in the bone marrow of *Mlkl^{S131P}* homozygotes could indicate a defect in hematopoiesis. To test this, we examined recovery from myelosuppressive irradiation as an assessment of hematopoietic function in mice carrying the *Mlkl^{S131P}* allele. Following myelosuppressive irradiation, recovery of hematopoietic cell numbers and circulating peripheral blood cells was significantly delayed in *Mlkl^{S131P/S131P}* mice compared with wild-type controls (Fig. 3a–f). In *Mlkl^{S131P/S131P}* mice, peripheral red blood cell and platelet numbers were significantly reduced at 14 days post irradiation, with the former also decreased at 21 days (Fig. 3a, b). Despite equivalent total white blood cell numbers at 21 days post irradiation, a significant reduction in peripheral monocyte numbers and a non-significant decrease trend in neutrophils was observed in *Mlkl^{S131P/S131P}* mice compared to wild-type controls (Fig. 3c, Supplementary Fig. 3A, B). These reductions in circulating blood cell numbers were accompanied by significant decreases in the nucleated viable, progenitor, and LSK cell populations in the bone marrow of *Mlkl^{S131P/S131P}* mice (Fig. 3d–f). Contrastingly, at 21 days post irradiation, *Mlkl^{S131P}* heterozygotes displayed a significant increase in their total nucleated viable cells in comparison to wild-type controls. This was driven predominantly by significant increases in the number of LSK cells (Fig. 3d, e). Impaired recovery of *Mlkl^{S131P/S131P}* LSK and progenitor cell numbers was characterized by increased expression of ROS and Annexin V at 21 days post irradiation (Supplementary Fig. 3C, D). *Mlkl^{S131P}* homozygotes had increased plasma G-CSF levels at 14 and 21 days post irradiation when compared to *Mlkl^{WT/WT}* controls (Fig. 3g). At 21 days, all other plasma cytokines, with exception of IL-1β and RANTES, were equivalent between genotypes (Supplementary Fig. 3E). *Mlkl^{WT/S131P}* and *Mlkl^{S131P/S131P}* mice had decreased plasma IL-1β levels, whilst *Mlkl^{S131P/S131P}* mice alone had reduced RANTES levels compared to *Mlkl^{WT/WT}* controls (Supplementary Fig. 3E).

To investigate whether the *Mlkl^{S131P}* homozygote defect was intrinsic to the hematopoietic stem cells, we performed competitive transplantation studies. *Mlkl^{WT/WT}*, *Mlkl^{WT/S131P}* or *Mlkl^{S131P/S131P}* bone marrow was mixed with GFP^+ competitor bone marrow in a 50:50 ratio and injected into Ly5.1 irradiated hosts. Six weeks post-transplantation, *Mlkl^{WT/WT}* bone marrow had competed with GFP^+ bone marrow effectively, whilst *Mlkl^{S131P/S131P}* bone marrow performed poorly, contributing to 7%, 5% and 9% of PBMCs (Ly5^+), red blood cells and platelets respectively (Fig. 3h, Supplementary Fig. 3F, G). Under competitive transplant conditions, *Mlkl^{WT/S131P}* bone marrow competed comparably to *Mlkl^{WT/WT}* with approximately 50% of the peripheral cells generated from these donor stem cells (Fig. 3h, Supplementary Fig. 3F, G). When mixed in excess at 70:30 with GFP^+ competitor bone marrow, *Mlkl^{S131P/S131P}* stem cells were again outcompeted contributing to <10% of the peripheral blood cells (Supplementary Fig. 3H–J). Thus, the defect in emergency hematopoiesis was intrinsic to *Mlkl^{S131P/S131P}* hematopoietic stem cells, and reminiscent of an intrinsic defect previously reported for the *Mlkl^{D139V}* autoactivating mutant[33].

## Recruitment of Ly6C^{hi} monocytes to sites of sterile inflammation is reduced in *Mlkl^{S131P/S131P}* mice

To examine if the defects in emergency hematopoiesis displayed by *Mlkl^{S131P/S131P}* mice could result in defective immune cell recruitment to sites of inflammation or infection, we employed a model of localized sterile inflammation induced by zymosan. Intra-peritoneal injection of zymosan results in a rapid influx of immune cells, predominantly neutrophils, into the peritoneal cavity over 72 hours[47]. The cellular content of the peritoneal cavity and numbers of circulating peripheral blood cells was examined at 4 and 24 hours post-zymosan injection. *Mlkl^{S131P/S131P}* mice displayed equivalent early recruitment of neutrophils and other immune cells to the peritoneum, with exception of reduced quantities of Ly6C^{hi} monocytes (Fig. 4a, b, Supplementary Fig. 4A–H). At 24 hours post-zymosan injection, this significant reduction in the recruitment of Ly6C^{hi} monocytes to the peritoneum of *Mlkl^{S131P/S131P}* mice was more pronounced (Fig. 4c, d, Supplementary Fig. 4I–P). In the peripheral blood, monocytes were significantly elevated in *Mlkl^{S131P/S131P}* mice in comparison to wild-type controls at 4 hours post-zymosan (Fig. 4e, Supplementary Fig. 4Q–S). However, by 24 hours, the number of peripheral blood monocytes was equivalent between genotypes, with only a non-significant increase in neutrophils observed in *Mlkl^{S131P/S131P}* mice (Fig. 4f, Supplementary Fig. 4T–V). *Mlkl^{WT/S131P}* mice exhibited increases in the peripheral blood lymphocyte and neutrophil numbers when compared to wild-type controls at 24 hours (Fig. 4f). We also measured cytokines present in the peritoneum at 4 and 24 hours post-zymosan injection (Fig. 4g, h, Supplementary Fig. 4W–Y). At 4 hours, *Mlkl^{S131P/S131P}* mice had increased quantities of IL-13 and MCP-1 when compared to wild-type controls (Fig. 4g, h). No statistically significant differences were observed in the quantities of peritoneal cytokines between any genotypes at 24 hours post-injection (Supplementary Fig. 4Y).

We previously observed that *Mlkl^{S131P/S131P}* cells exhibited reduced response to TSI-induced necroptosis. To assess whether this defect in necroptotic function was observed in activated immune cells recruited to sites of inflammation, we isolated neutrophils from the peritoneum at 4 hours post-zymosan injection. Consistent with our previous in vitro findings, no differences were observed in the capacity of *Mlkl^{S131P/S131P}* inflammatory neutrophils to undergo spontaneous apoptosis (Supplementary Fig. 4Z). However, the percentage of *Mlkl^{S131P/S131P}* inflammatory neutrophils dying following necroptotic stimulation with TSI was decreased in comparison to wild-type neutrophils (Fig. 4i).

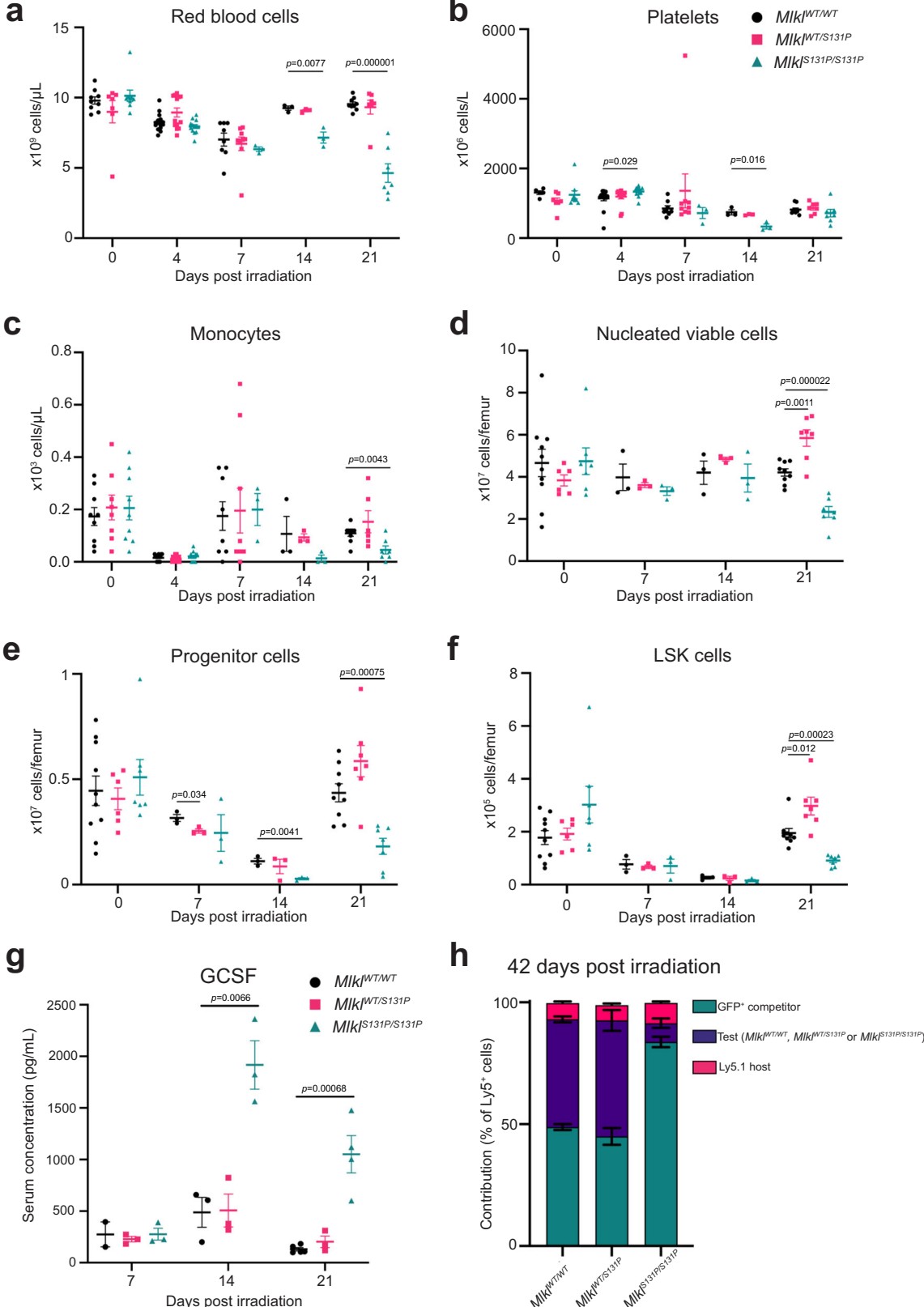

***Mlkl^S131P* homozygotes exhibit reduced capacity to clear *Salmonella* infection**

Reductions in the recruitment of Ly6C^hi monocytes to sites of inflammation raise important questions as to whether the *Mlkl^S131P* mutation impacts defense against pathogens. *Salmonella enterica* has long been an important pathogenic selective pressure in humans[48]. For non-typhoidal *Salmonella* serovars, such as *S. enterica* serovar *Typhimurium*, human infection is typically limited to the gastric mucosa[49,50]. In mice and immunocompromised humans, *S. Typhimurium* can cause severe systemic disease after dispersal from the gut by dendritic cells and macrophages to peripheral organs including the spleen and liver[51].

**Fig. 3 | *Mlkl*^S131P^ mice show delayed recovery from myelosuppressive irradiation. a** Peripheral red blood cells (*n* = 9, 8, 8, 15, 13, 13, 8, 9, 3, 3, 3, 3, 9, 9, 7, 7), **b** platelets (*n* = 5, 7, 9, 15, 13, 13, 8, 9, 3, 3, 3, 3, 9, 9, 7, 7) and **c** monocytes (*n* = 9, 8, 9, 15, 13, 13, 8, 9, 3, 3, 3, 3, 9, 6, 7) in *Mlkl*^WT/WT^, *Mlkl*^WT/S131P^, and *Mlkl*^S131P/S131P^ mice following treatment with 5.5 Gy radiation. Quantified nucleated viable cells (**d**), progenitor (**e**), and LSK (**f**) populations in the bone marrow of mice after myelosuppressive radiation (**d**–**f** *n* = 10, 6, 7, 3, 3, 3, 3, 3, 9, 7, 7). **g** Multiplex measurement of plasma G-CSF levels at 7, 14 and 21 days post-myelosuppressive radiation. Each symbol represents one individual mouse sampled, with mean ± SEM of *n* = 2, 3, 3, 3, 3, 3, 5, 3, 4

biologically independent mice from two separate experiments. Bone marrow from *Mlkl*^WT/WT^, *Mlkl*^WT/S131P^ or *Mlkl*^S131P/S131P^ mice on CD45^Ly5.2^ background was mixed with wild-type GFP⁺ competitor bone marrow on a CD45^Ly5.2^ background and transplanted into irradiated CD45^Ly5.1^ recipients. **h** Relative donor contribution to PBMCs was assessed at 6 weeks post-transplantation. Mean ± SEM shown for *n* = 11, 6, 9 biologically independent recipient mice. Host contribution (CD45^Ly5.1^) depicted in pink, GFP competitor in green and test (*Mlkl*^WT/WT^, *Mlkl*^WT/S131P^ or *Mlkl*^S131P/S131P^) in purple. *P* value calculated using an unpaired, two-tailed Students *t* test (**a**–**g**). Source data are provided as a Source Data file.

*Mlkl*^S131P^ homozygote, heterozygote and wild-type mice were infected with a metabolically growth-attenuated *Salmonella Typhimurium* strain BRD509 (from here referred to as *Salmonella)*, via oral gavage (1 × 10⁷ colony forming units (CFU)). Daily monitoring for the 14 day infection period showed no differences in core body temperature or body mass between genotypes (Supplementary Fig. 5A, B). Consistent with normal weights, there were no obvious differences in the integrity of the intestinal epithelial barrier or intestinal monocyte and macrophage populations at infection endpoint (Fig. 5a). Despite this, *Mlkl*^S131P/S131P^ mice had increased *Salmonella* burden in both the spleen and liver compared to wild-type controls (Fig. 5b, Supplementary Fig. 5C, D). Bacterial colonization in the feces was increased in female *Mlkl*^S131P/S131P^ mice relative to wild-type female controls (Fig. 5b, Supplementary Fig. 5E). In the peripheral blood, *Mlkl*^S131P/S131P^ mice had significantly reduced numbers of circulating lymphocytes and monocytes, as well as a trend towards decreased numbers of circulating neutrophils (Fig. 5c, Supplementary Fig. 5F–H). Infected *Mlkl*^S131P/S131P^ mice also exhibited significant decreases in the quantity of splenic Ly6C^hi^ monocytes in comparison to infected wild-type controls (Fig. 5d, e, Supplementary Fig. 5I–O). At 14 days post-infection, *Mlkl*^WT/S131P^ and *Mlkl*^S131P/S131P^ mice both had significant increases in plasma concentrations of MCP-1, when compared to wild-type controls (Fig. 5f). All other plasma cytokines were equivalent between genotypes at 14 days post-infection (Fig. 5g).

We did not observe any differences in the kinetics of *Salmonella*-induced cell death in ex vivo BMDM infection assays, nor inflammasome activation, measured by GSDMD cleavage (Supplementary Fig. 5P, Q). Together these data show a hindered pathogen defense in *Mlkl*^S131P^ homozygotes accompanied by widespread immunophenotypic deficiencies. Combined, our investigations of *Mlkl*^S131P^ homozygotes under challenge highlight a defect in emergency hematopoiesis that manifests in a disruption to integral inflammatory and immune responses. This provides important insights into the potential modulation of immunoinflammatory disorders in *MLKL S132P* carriers.

## Discussion

A high frequency *MLKL S132P* polymorphism present in 2–3% of the global population confers a gain-of-function to MLKL resulting in hematopoietic dysfunction and immune cell defects in a genetically modified mouse model. In human cells, MLKL^S132P^ was resistant to chemical inhibition by necrosulfonamide treatment and endogenous inhibitory phosphorylation at Serine 83. A gain in necroptotic function was also observed when the murine equivalent, *Mlkl*^S131P^, was examined in situ. Fibroblasts isolated from *Mlkl*^S131P^ homozygotes exhibit RIPK3-independent cell death in the presence of IFN-β, a strong inducer of *Mlkl* gene expression. Under regular culture conditions, MLKL^S131P^ protein levels are reduced relative to MLKL^WT^, manifesting in a reduced capacity for necroptosis in fibroblasts and immune cells stimulated with TSI. While the reduction in MLKL^S132P^ protein levels was also evident in PBMCs isolated from one human individual heterozygous for this polymorphism, our capacity to fully compare between mouse and human systems is limited by the lack of suitable human cell lines that express *MLKL*^S132P^ from its endogenous gene locus. The low number of

patients identified in our study limits our current conclusions regarding the disease implications of the *MLKL S132P* polymorphism however, the contribution of *MLKL S132P* to human inflammatory disease remains a tantalizing and important avenue for future study.

Extensive characterization of the *Mlkl*^S131P^ mouse unveiled steady state decreases in the bone marrow pool of inflammatory Ly6C^hi^ monocytes, that were also reflected in reduced numbers at sites of sterile inflammation and bacterial infection. During zymosan-induced peritonitis, increased peripheral monocytes were observed at 4 hours however, Ly6C^hi^ monocytes were significantly reduced in the peritoneum of homozygotes at 24 hours. Under infection challenge, an increased burden of *Salmonella* was present in the spleen and liver of homozygotes, accompanied by significant reductions in the circulating lymphocytes and monocytes. In both cases, monocyte deficits were accompanied by an increase in plasma MCP-1 levels. These deficits are not attributed to any inherent variations in *Mlkl*^S131P^ mouse monocyte/macrophage progenitor cell populations at steady state however, whether these deficits are the result of Ly6C^hi^ monocyte recruitment to an unidentified site of inflammation in *Mlkl*^S131P/S131P^ mice at steady state remains an important avenue of open investigation. Interestingly, *Mlkl*^S131P^ stem cells did exhibit an increased capacity to form macrophage colonies ex vivo under IL-3, EPO and SCF combination treatment. These data led us to hypothesize that *Mlkl*^S131P/S131P^ Ly6C^hi^ monocytes may be more prone to differentiation than their *Mlkl*^WT/WT^ counterparts under specific conditions. Hematopoietic dysfunction in *Mlkl*^S131P^ homozygotes is evident following radio-ablation, with a severely reduced capacity for stem cells to repopulate in situ, or even *Mlkl*^WT^ recipients. Hematopoietic stem cells expressing the constitutively active *Mlkl*^S131P^ are characterized by increased ROS and Annexin V positivity following irradiation, marking their enhanced propensity for cell death. Taken together, the capacity for monocyte generation, activation and peripheral differentiation within the *Mlkl*^S131P^ mouse under conditions of stress has emerged as the most enticing area for future investigation.

Prior to this work two other single point mutant *Mlkl* mouse models, *Mlkl*^D139V^ and *Mlkl*^S83G^, had been reported, both of which exhibited full or partial homozygous postnatal lethality characterized by severe inflammation[29,33]. In stark contrast to these models, the *Mlkl*^S131P^ homozygotes were born normal and with no signs of inflammatory disease at steady state. However, upon challenge, similarities between *Mlkl*^S131P^ and *Mlkl*^D139V^ mice were revealed. Hematopoietic dysfunction is observed following myelosuppressive irradiation in homozygous or heterozygous mice harboring the *Mlkl*^S131P^ or *Mlkl*^D139V^ allele, respectively[33].

Despite the ostensible basal state phenotypes of *Mlkl*^D139V^ and recently described *Mlkl*^S83G^ homozygotes differing from the *Mlkl*^S131P^ homozygotes[29,33], there were similarities observed at the molecular level. As for MLKL^S131P^, RIPK3-independent gain-of-function was reported for MLKL^D139V^, but not MLKL^S83G[29,33]^. This major similarity in molecular function is not unexpected considering the proximity of the D139 and S131 residues within mouse MLKL. Constitutive activity of the D139V brace helix mutants is restrained by cellular mechanisms that limited protein level, with decreases in endogenous MLKL observed in cells generated from *Mlkl*^D139V^ homozygotes. Interestingly, ubiquitin-

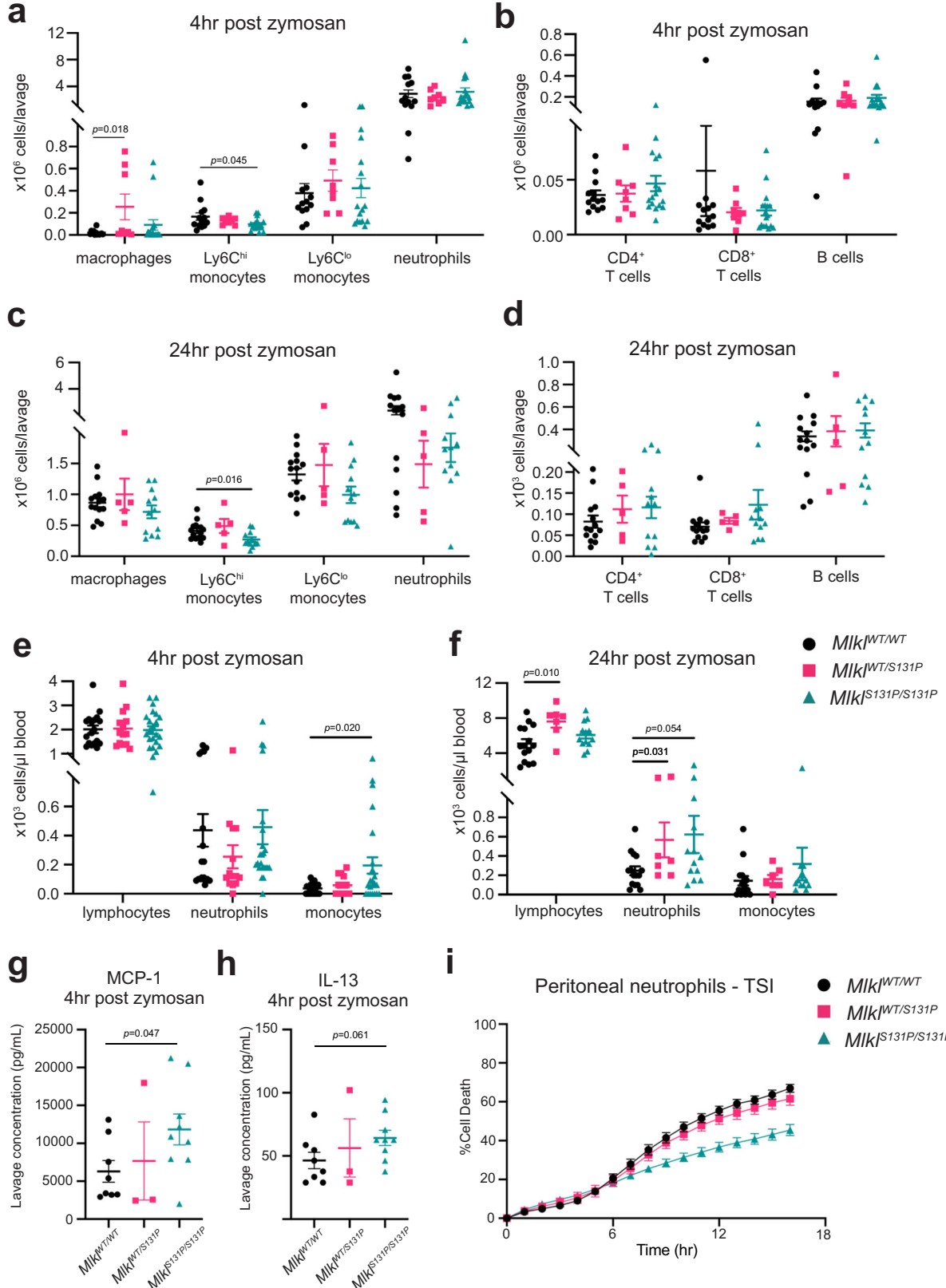

mediated targeting of MLKL to the lysosome was identified to be mechanistically involved in the restraint of necroptosis in cells expressing MLKL$^{D139V}$, and as a means of clearing endosomal *Listeria monocytogenes* and *Yersinia enterocolita*[33,52,53]. Exploring if the reduction in cellular MLKL levels we observed in the PBMCs of a human carrier of the *MLKL$^{S132P}$* polymorphism, or cells isolated from the

*MLKL$^{S131P}$* mouse, is likewise mediated by ubiquitination and lysosomal targeting will be an important next step, particularly as it relates to the clearance of these intracellular bacteria from human cells. Ablation of the inhibitory phosphorylation site in *Mlkl$^{S83G}$* mice similarly subjects MLKL to downregulation in mouse cells[29]. In sum, our analysis of MLKL$^{S131P}$ further highlights the importance of cellular mechanisms

**Fig. 4 | *Mlkl^S131P* recruited inflammatory neutrophils are less sensitive to TNF-induced necroptosis. a–d** Flow cytometry quantification of peritoneal innate (macrophages, Ly6C^hi monocytes, Ly6C^lo monocytes and neutrophils) and adaptive (CD4^+ T cells, CD8^+ T cells and B cells) immune cells at **a, b** 4- (n = 13, 8, 17) or (**c, d**) 24- (n = 14, 5, 12) hours post zymosan injection. ADVIA quantification of circulating immune cells (lymphocytes, neutrophils, and monocytes) at **e** 4- (n = 18, 14, 23) and **f** 24- (n = 15, 7, 13) hours post zymosan injection. Multiplex measurement of MCP-1 (**g**) and IL-13 (**h**) cytokine levels in peritoneal lavage at 4 hours post-zymosan injection (n = 8, 3, 9). **a–g** Error bars represent mean ± SEM with each symbol representing an independent mouse. Mice from the 4- and 24-hour time points were pooled from 3 and 2 independent experiments respectively. **i** Neutrophils isolated from the peritoneum 4 hours post-zymosan injection were treated with necroptotic stimulus (TNF, Smac mimetic, IDN-6556; TSI) for 16 hours and SYTOX Green positive (dead) cells quantified using IncuCyte SX5 live cell imaging. Data were collected from one independent experiment with male and female data pooled, error bars indicate mean ± SEM of n = 13 Mlkl^WT/WT, 6 Mlkl^WT/S131P, 14 Mlkl^S131P/S131P individual mice. *P* value calculated using an unpaired, two-tailed Students t test. Source data are provided as a Source Data file.

that clear activated MLKL from cells below a threshold to reduce aberrant necroptotic cell death.

The finding that MLKL^S132P retains necroptotic killing activity despite simulated inhibitory phosphorylation at Ser83 in human cells is highly notable. The inflammatory phenotypes that result from homozygosity of the phosphoablating (S83G) mutation provides insight into the potential disease development that may occur in carriers of the *MLKL S132P* polymorphism under environmental and cellular scenarios where Ser83 inhibitory phosphorylation is deployed. Identifying the kinase(s) responsible for phosphorylation at MLKL Ser83 and understanding which cell types are primed to deploy this kinase(s) is an important next step in determining if this polymorphism promotes clinically relevant changes to homeostasis or disease outcomes in humans.

Evidence for positive selection has been found for over 300 immune-related gene loci and many of these have been found to be associated with the incidence of autoimmune and autoinflammatory disease in modern humans[54,55]. Many of these variants have also been mechanistically linked to pathogen defense (Karlsson et al., 2014, Ramos et al., 2015), with pathogenic microbes a major driver of genetic selection over human history. While a diminished capacity for mice to clear disseminated *Salmonella* would argue against the hypothesis that the *MLKL S132P* polymorphism has been positively selected for in human populations, it is important to note that only 1.4% of human carriers are homozygotes[39]. *MLKL S132P* heterozygosity is by far the most prevalent, and evolutionarily relevant, human scenario. *Mlkl^S131P* heterozygote mice displayed gene dosage phenotypes consistent with homozygotes, with one exception. Following sublethal myelosuppressive irradiation, a gain in bone marrow hematopoietic stem cell numbers in heterozygotes was observed at day 21 whilst a catastrophic drop occurred in homozygotes. This increased stem cell capacity does not persist long-term. In competitive bone marrow transplants, *Mlkl^WT/S131P* stem cells compete similarly to *Mlkl^WT/WT* stem cells at 6 weeks post-transplantation. However, the increased fitness conferred to heterozygous mice at this early timepoint provides intriguing insights into other selective pressures that may have promoted accumulation of this polymorphism in humans[56]. Sepsis, where emergency hematopoiesis is an essential determinant in survival, is an interesting and highly evolutionarily important avenue of exploration for the study of *MLKL S132P* polymorphism frequency[57,58]. Since nutrition is another important driver of genetic adaptation in humans, exploring the role of *MLKL S132P* in metabolic disease is also of interest. An important precedent for this is a recent report of an association between human RIPK1 promoter polymorphisms and diet-induced obesity[59]. The potential for negative selection of this polymorphism over time is also an important avenue of exploration in light of the recent discovery of a common *TYK2* variant and its role in enhancing susceptibility to severe infection by historically important human pathogen mycobacterium tuberculosis[60,61].

By many, MLKL is viewed as a potential therapeutic target for drug discovery due to its established involvement in multiple human diseases, especially inflammatory pathologies[33,62–64]. To date no human clinical trials have been conducted on MLKL-targeted small molecules, although several inhibitors have been reported in the literature, including human MLKL inhibitor NSA, which all function by targeting Cys86[16,65]. Intriguingly, we show that in the presence of NSA human MLKL^S132P protein displays a gain-of-function that results in non-inhibitable cell death. While NSA has been a fundamental tool for in vitro studies of necroptosis, our findings raise interesting questions about the suitability of Cys86-targeted MLKL inhibitors for the 2–3% of the population that carry the *MLKL S132P* polymorphism.

## Methods

### Patient recruitment, ethics, and informed consent

Patients and their relatives were recruited from the Department of Clinical Immunology and Allergy, Royal Melbourne Hospital, Victoria, Australia, and the Centre for Personalized Immunology, Australian National University, Canberra, Australia. Unrelated, age and sex matched healthy controls that did not carry the *MLKL p.Ser132Pro* polymorphism were recruited via the Volunteer Blood Donor Registry, Parkville. Patient #1 is a (self-reported) female, aged in their 30's, diagnosed with SAPHO syndrome and patient #2 is a (self-reported) female, aged in their 50's, diagnosed with systemic IgG4 disease. Healthy control was a (self-reported) female that was age matched to patient #2. Selection of patients from this registry was based upon genotype and their geographical location (Royal Melbourne Hospital), and no other patient characteristics, including sex, was considered. No suitable male carriers of the *p.Ser132Pro* polymorphism were available in this study. Informed written consent was obtained from all participants for genomic analysis and immunological studies. All procedures were performed and are reported here with the approval of human ethics review boards of all Institutes that participated in human genetics studies; Australian National University, The Walter and Eliza Hall Institute of Medical Research (approved projects 2009.162, 10/02) and with the 1964 Helsinki declaration and its later amendments or comparable ethical standards.

### Genomic analysis

Whole exome sequencing was performed by the Canberra Clinical Genomics service. Libraries were prepared and enriched using the SureSelect Clinical Research Exome v2 kit (Agilent Technologies), and targeted regions were sequenced using an Illumina sequencing system with 100 bp paired-end reads, with an average depth of >35. Reads were aligned to the human genome reference sequence (GRCh38) using the Burrows-Wheeler Aligner (BWA-MEM), and variant calls made using the Genomic Analysis Tool Kit (GATK). Patient #1 and #2 *MLKL* sequence data can be directly accessed from the Sequence Read Archive (SRA BioProject accession number PRJNA1007397) via the following link; https://www.ncbi.nlm.nih.gov/sra/?term=PRJNA1007397.

### Animal ethics

All experiments were approved by the WEHI Animal Ethics Committee (2020.027) in accordance with the Prevention of Cruelty to Animals Act (1986) and the Australian National Health and Medical Research Council Code of Practice for the Care and Use of Animals for Scientific Purposes (1997).

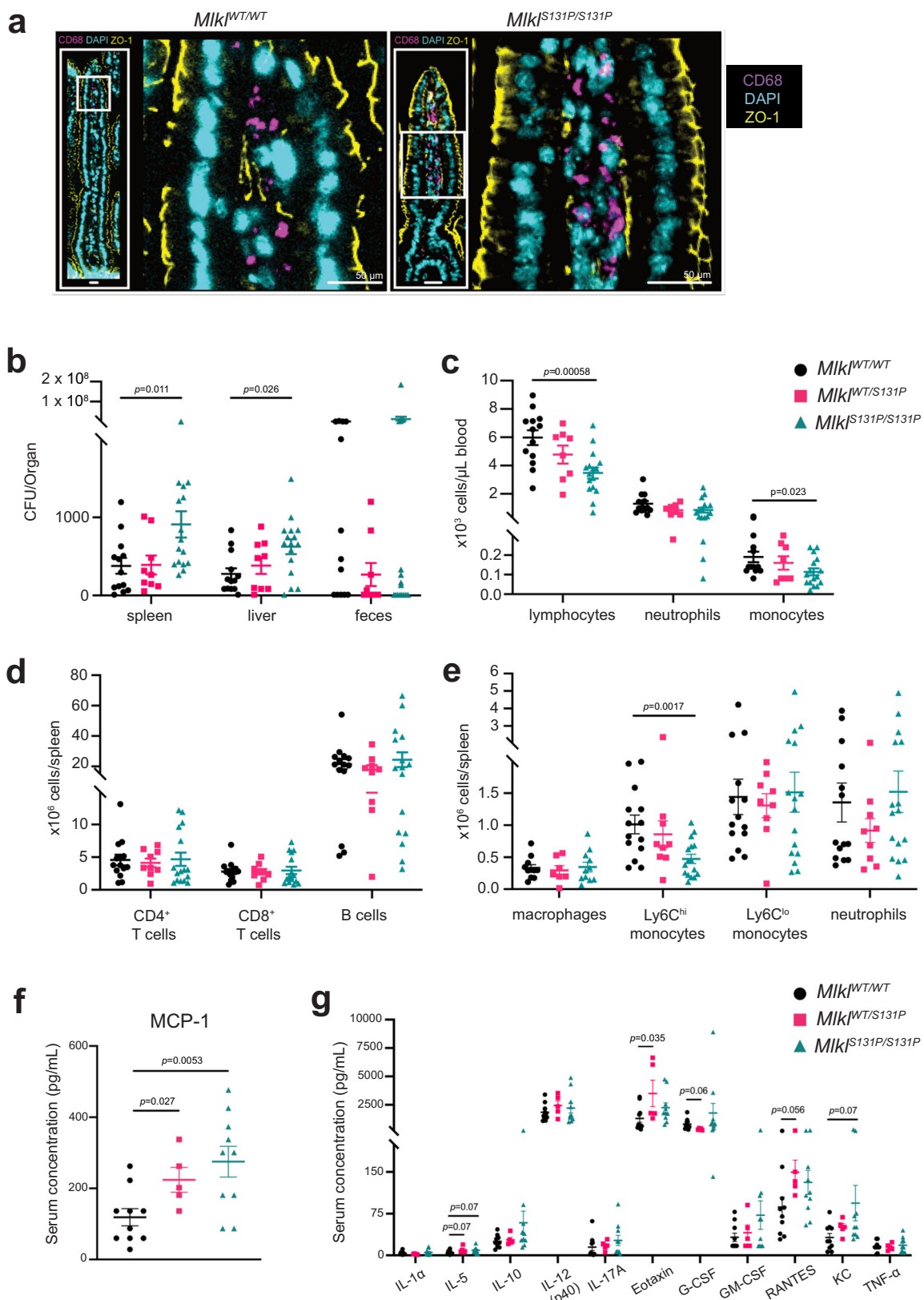

## Mice

*Mlkl*[S131P] CRISPR mouse strain[33] was bred and maintained at WEHI (Parkville) Animal Facility. On limited occasions, C57BL/6 J mice bred at WEHI Kew facility were used as non-littermate WT controls (where indicated). *C57BL/6-CD45*[Ly5.1] mice were bred and imported from WEHI Kew facility. These are temperature (21 °C ± 3 °C) and humidity (40–70%) controlled specific pathogen free facilities with a 14 h:10 h day:night cycle. All mice were euthanised by $CO_2$ asphyxiation utilizing Quietek automated $CO_2$ induction units. Experiments were performed with both male and female *Mlkl*[S131P] colony mice aged from 6–52 weeks old (specific age ranges denoted in experimental methods). Data for both sexes were presented combined or separately as indicated.

**Fig. 5 | *Mlkl^{S131P}* homozygote mice exhibit bacterial clearance defects following oral *Salmonella* infection. a** ZO-1 (yellow) and CD68 (purple) staining in epithelial barrier of intestinal sections taken at 14 days post-*Salmonella* infection. Images are representative of *n* = 5 biologically independent mice per genotype. **b** Increased bacterial burden observed in spleen and liver, but not feces in *Mlkl^{S131P/S131P}* mice at infection endpoint (*n* = 13, 9, 16). **c** Quantification of circulating white blood cells (lymphocytes, neutrophils and monocytes) using ADVIA hematology (*n* = 13, 8, 16) and (**d**, **e**) splenic adaptive (CD4^+ T cells, CD8^+ T cells and B cells) and innate (macrophages, Ly6C^hi monocytes, Ly6C^lo monocytes and neutrophils) immune cells using flow cytometry (*n* = 14, 9, 16; with exception of macrophages *n* = 11, 7, 11). **f**, **g** Multiplex measurement of plasma cytokine levels at 14-days post *Salmonella* infection (*n* = 10, 5, 10). *Salmonella* infection was performed on three independent occasions. Error bars represent mean ± SEM with each symbol representing an individual mouse. *P* value calculated using a two-tailed Mann–Whitney test **b** or an unpaired, two-tailed Students *t* test (**c**–**g**). Source data are provided as a Source Data file.

## Reagents

Antibodies; Rat anti-mMLKL 8F6[33](1:2000), rat anti-mMLKL 5A6[66] (1:1000; available from Merck-Millipore as MABC1634), rat anti-hMLKL 10C2[22] (1:1000; available from Merck-Millipore as MABC1635), rat anti-hMLKL 7G2[22] (1:1000; available from Merck-Millipore as MABC1636), rat anti-hRIPK3 1H2[4] (1:1000; available from Merck-Millipore as MABC1640) were produced in-house. Mouse anti-actin (A-1978; 1:5000) was purchased from Sigma-Aldrich, rabbit anti-GAPDH (#2118; 1:2000–5000) was purchased from Cell Signalling Technology, rabbit anti-VDAC (AB10527; 1:10000) was purchased from Millipore, rabbit anti-human pMLKL (EPR9514; 1:1000–3000) was purchased from Abcam, and rabbit anti-mouse pMLKL (D6E3G; 1:1000) was purchased from Cell Signalling Technology, rabbit anti-GSDMD (AB209845; 1:1000) was purchased from Abcam, and rabbit anti-hRIPK1 (D94C12; 1:1000) was purchased from Cell Signalling Technology. Western blot antibody details are outlined in Supplementary Table 1.

Cell treatments were completed with agonists/antagonists at the following concentrations: 100 ng/ml recombinant hTNF-Fc (produced in house as in[67]), 500 nM Smac mimetic Compound A (provided by Tetralogic Pharmaceuticals; as in[68], 5 µM Pan-caspase inhibitor IDN-6556 (provided by Tetralogic Pharmaceuticals), 1 µM NSA (Merck #480073), 50 µM Necrostatin 1 S (Nec-1s; Merk #504297), 2 µM GSK' 872 (SynKinase #SYN-5481), 10–20 ng/ml lipopolysaccharide (LPS; Sigma #L2630), 25 µg/ml polyinosinic:polycytidylic (Poly I:C; Novus), 200 nM MG132 (Merck #474790), 2 nM PS341 (Sigma #504314), and 30 ng/ml mouse IFNβ (R&D Systems #8234-MB-010)

## Cell lines

Mutations were introduced into a human *MLKL* DNA template (from DNA2.0, CA) using oligonucleotide-directed PCR and sub-cloned into the pF TRE3G PGK puro vector[18]. Vector DNA was co-transfected into HEK293T cells with pVSVg and pCMV δR8.2 helper plasmids to generate lentiviral particles. U937 (WT and *MLKL^{−/−}*) and HT29 (WT and *MLKL^{−/−}*) were then stably transduced with exogenous human *MLKL* ligated into pFTRE3G. Successfully transduced cells were selected using puromycin (2.5 µg/mL; StemCell Technologies). The following oligonucleotides were used for the assembly of constructs:

*hMLKL^{S132P}* fwd; 5'-GCCAAGGAGCGCCCTGGGCACAG-3'
*hMLKL^{S132P}* rev; 5'-CTGTGCCCAGGGCGCTCCTTGGC-3'
*hMLKL^{TSEE}* fwd; 5'-GAGGAAAACACAGGAGGAAATGAGTTTGGGAAC-3'
*hMLKL^{TSEE}* rev; 5'-GTTCCCAAACTCATTTCCTCCTGTGTTTCCTC-3'
*hMLKL^{S83A}* fwd; 5'-GTTCAGCAATAGAGCCAATATCTGCAG-3'
*hMLKL^{S83A}* rev; 5'-CCTGCAGATATTGGCTCTATTGCTGAAC-3'
*hMLKL^{S83D}* fwd; 5'-GAAAAGTTCAGCAATAGAGACAATATCTGCAGGTTTC-3'
*hMLKL^{S83D}* rev; 5'-GAAACCTGCAGATATTGTCTCTATTGCTGAACTTTTC-3'
*hMLKL^{S83C}* fwd; 5' GTTCAGCAATAGATgCAATATCTGCAGG-3'
*hMLKL^{S83C}* rev; 5' CCTGCAGATATTGcATCTATTGCTGAAC-3'
*hMLKL^{R82S}* fwd; 5' GAAAAGTTCAGCAATAGCTCCAATATCTGCAG-3'
*hMLKL^{R82S}* rev; 5' CTGCAGATATTGGAGCTATTGCTGAACTTTTC-3'

Primary mouse dermal fibroblasts (MDF) were prepared from skin taken from the head and body of E19.5 male and female mice. These MDFs were immortalized by stable lentiviral transduction with SV40 large T antigen. Bone marrow-derived macrophages (BMDM) were generated from the long bones of adult mice and grown for 7 days in DMEM supplemented with 15% L929 cell supernatant.

Human blood (patient and healthy donor) was collected by collaborators via venipuncture at the Royal Melbourne Hospital. Collected blood was diluted with PBS and layered on an equal volume of Histopaque (density 1.077 g/ml) and centrifuged for 30 minutes, 700 × *g* at 20 °C. The layer containing peripheral blood mononuclear cells (PBMC) was harvested and washed with PBS, then frozen in FCS + 10% DMSO and stored in liquid nitrogen.

CRISPR-edited derivative HT29 and U937 cells lacking MLKL were reported previously[45] and were produced in-house using unmodified HT29 cells supplied by Mark Hampton (originally from ATCC) and U937 cells supplied by ATCC. HEK293T cells were sourced from the laboratory of John Silke and were originally purchased from the ATCC.

## Culture of cell lines

Primary MDFs, immortalised MDFs and HT29s were cultured in DMEM + 8% FCS. BMDMs were cultured in DMEM + 15% FCS + 20% L929. U937 and PBMCs were cultured in RPMI + 8% FCS. All cell lines were grown at 37 °C and 10% (v/v) CO_2.

## Western blot

U937 cells were seeded into 48-well plates at 60,000 cells/well and induced for 3 hours with doxycycline (20 ng/mL, 100 ng/mL or 500 ng/mL) to stimulate *MLKL* expression. HT29 cells were seeded into 48-well plates at 45,000 cells/well and following 12–14 hours of cell adherence, cells were induced overnight with doxycycline (20 ng/mL, 100 ng/mL or 500 ng/mL) for stimulation of *MLKL* expression. BMDMs were plated at 400,000 cells/well in a 24-well plate and MDFs (primary and immortalised) were plated at 30,000 cells/well in a 48-well plate. Cells were stimulated as indicated for 6 hours, except for BMDMs stimulated with LPS for 2 hours before addition of Smac mimetic Compound A for a further 4 hours. Human primary PBMCs were plated at 45,000 cells/well and stimulated for 4 hours. All cells were harvested in SDS Laemmli's lysis buffer, boiled at 100 °C for 10–15 min, and then resolved by 4–15% Tris-Glycine SDS-PAGE (Bio-Rad). Proteins were transferred to nitrocellulose or PVDF membrane and probed with antibodies as indicated.

## BN-PAGE

Immortalised MDFs were plated at 1 × 10^6 cells/well in a 6-well plate, left to settle for 4 hours and stimulated with TNF + IFN-β for 22 hours or TSI for 6 hours. *MLKL^{−/−}* HT29 cells were plated at 0.25 × 10^6 cells/well in a 12-well plate and left to settle overnight. HT29 cells were then stimulated with 100 ng/ml doxycycline overnight followed by TSI or TSI + NSA treatment for 2, 4 or 6 hours. All cells were permeabilised in MELB buffer (20 mM HEPES pH 7.5, 100 mM sucrose, 100 mM KCl, 2.5 mM MgCl_2. 2 µM N-ethyl maleimide) supplemented with 0.025% digitonin, protease and phosphatase inhibitors. Using centrifugation (5 min, 11,000 × *g*), cells were fractionated into cytoplasmic and crude membrane fractions and solubilised in 1% digitonin. Samples were mixed with BN-PAGE loading buffer and resolved using 4–16% NativePAGE (Invitrogen) gel. Proteins were transferred to PVDF membrane that was subsequently destained (50% v/v Methanol, 25% v/v acetic

acid), denatured (6 M Guanidine hydrochloride, 5 mM β-mercaptoethanol, 10 mM Tris pH 6.8), blocked (1% BSA or 2% skim milk) and probed with denoted antibodies. Samples of fractions were reduced in SDS Laemmli's lysis buffer and then immunoblotted for loading controls.

## IncuCyte analysis

Primary MDFs were plated at 8000 cells per well in a 96-well plate. BMDMs were plated at 150,000 cells per well on day 6 of culture in a 48-well plate. Primary MDFs and BMDMs were left to settle overnight before stimulation. The next day MDFs and BMDMs were stimulated in culture media supplemented with propidium iodide. Immortalised MDFs were plated at 10,000 cells/well in a 96-well plate and left to settle for 4–6 hours. Cells were then stimulated in DMEM + 1% FCS supplemented with SYTOX Green (1:10,000). HT29 cells were plated at 45,000 cells per well in a 48-well plate and left to settle for 6 hours before overnight doxycycline pre-stimulation (20 ng/mL, 100 ng/mL or 500 ng/mL). HT29 cells were stimulated in Phenol Red-free media supplemented with 2% FCS, 1 mM Na pyruvate, 1 mM L-GlutaMAX, SYTOX Green (Invitrogen, S7020) and either DRAQ5 (Thermofisher, #62251) ($MLKL^{-/-}$ HT29) or SPY_620 (Spirochrome, SC401) (WT HT29).

U937 cells were plated at 60,000 cells per well in a 48-well plate and were induced with doxycycline (20 ng/mL, 100 ng/mL or 500 ng/mL) for 3 hours. Cells were then stimulated in Phenol Red-free media supplemented with 2% FCS, 1 mM Na pyruvate, 1 mM L-GlutaMAX, SYTOX Green and either DRAQ5 ($MLKL^{-/-}$ HT29) or SPY_620 (WT HT29).

Neutrophils isolated from the peritoneum at 4 hours post-zymosan injection were counted and plated at 60,000 cells/well in a 48-well plate. Plating media (RPMI + 8% FCS) was supplemented with SYTOX Green and DRAQ5 dyes.

Images were taken every hour using IncuCyte SX5 or S3 imager and analysed using Sartorius Incucyte Imager Software v2022B, v2021B, v2020C & v2018A. Percentage values were quantified by number of dead cells (SYTOX Green or propidium iodide positive) as a proportion of total cell number (DRAQ5 or SPY620 positive).

## TNF ELISA

100,000 cells were stimulated with LPS (10 ng/mL) or Poly I:C (2.5 μg/mL) for 3 hours. PBMC supernatant cytokine content was measured by ELISA (R&D: STA00C) according to the manufacturer's instructions. The measurements were performed in technical triplicates.

## Mouse histopathology

Organs from male and female $Mlkl^{S131P}$ mice between 7–9-months of age were fixed in 10% buffered formalin and examined by histopathologists and veterinary histopathologists at the Australian Phenomics Network, Melbourne.

## Hematological analysis

Cardiac, submandibular or retro-orbital blood collected from male and female $Mlkl^{S131P}$ colony mice at steady state (8–52 weeks old) or following challenge was placed into EDTA-coated tubes. Blood cells were left undiluted or diluted 2- to 11-fold in DPBS for automated blood cell quantification using an ADVIA 2120i hematological analyser on the same day as harvest.

## Cytokine quantification

All plasma was collected by centrifugation ($10,000 \times g$, 5 min) and stored at −80 °C. Lavage fluid and plasma cytokine quantities were measured by Bioplex Pro mouse cytokine 23-plex assay (Bio-Rad #M60009RDPD) according to manufacturer's instructions. When samples were denoted as <OOR, below reference range, for a particular cytokine they were assigned the lowest recorded value for that cytokine across all samples. When samples were denoted as >OOR, above

reference range, for a particular cytokine they were assigned the highest recorded value for that cytokine across all samples.

## Colony forming assays

Bone marrow from male and female $Mlkl^{S131P}$ colony mice was prepared as a single cell solution in balanced salt solution (0.15 M NaCl, 4 mM KCl, 2 mM $CaCl_2$, 1 mM $MgSO_4$, 1 mM $KH_2PO_4$, 0.8 mM $K_2HPO_4$, and 15 mM N-2-hydroxyethylpiperazine-N'2-ethane-sulfonic acid supplemented with 2% [v/v] bovine calf plasma). Clonal analysis of bone marrow cells ($2.5 \times 10^4$) was performed in 1 mL semi-solid agar cultures of 0.3% agar in DMEM containing 20% newborn calf plasma, stem cell factor (SCF; 100 ng/mL; in-house), erythropoietin (EPO; 2 U/mL; Janssen), interleukin-3 (IL-3; 10 ng/mL; in-house), G-CSF ($10^3$ U/mL; PeproTech), granulocyte-macrophage colony stimulating factor (M-CSF; $10^3$ U/mL; in-house). Cultures were incubated at 37 °C for 7 days in a fully humidified atmosphere of 10% $CO_2$ in air, then fixed, dried onto glass slides, and stained for acetylcholinesterase, luxol fast blue, haematoxylin, and the number and type of colonies were determined, blinded.

## Mouse model of Salmonella infection

Male and female $Mlkl^{S131P}$ colony mice used in this experiment were a mix of littermates and non-littermates aged 6–12 weeks and wild-type mice that were littermates behaved equivalently to non-littermates. Mice were infected with Salmonella enterica serovar Typhimurium strain BRD509[69] at $10^7$ colony forming units (CFU) by oral gavage. Mice were euthanized by $CO_2$ asphyxiation at 14 days post-infection. Cardiac bleeds were taken, and blood populations analyzed using an ADVIA hematology analyser. Liver, spleen, and feces were harvested for enumeration of viable bacteria on nutrient agar. Organs from infected mice were weighed and homogenized in 2 mL (spleens), 5 mL (livers) or 1 mL (feces) of PBS. Homogenates were serially diluted (in duplicate) in PBS and 10 μl drops plated out in duplicate onto LB agar (+ streptomycin) and incubated overnight at 37 °C. CFU/mL was calculated per organ for each mouse and then standardized to CFU/organ based upon organ weight. A portion of the spleen was processed for flow cytometry analysis.

## In vitro Salmonella infection

Salmonella Typhimurium strain SL1344 was grown shaking at 37 °C overnight in LB broth supplemented with 50 μg/mL streptomycin. $OD_{600}$ was determined using a spectrophotometer to calculate multiplicity of infection (MOI). BMDMs on day 6 of differentiation were plated at $4 \times 10^5$ cells/well in a 6-well plate for western blot analyses or $5 \times 10^4$ cells/well in a 96-well plate for LDH assays. Cells were infected at MOI:25 for western blot or MOI:10 or 50 for LDH assays in antibiotic- and serum-free DMEM for denoted incubation times. For all experimental analyses, following 30 minute incubation, cells were washed and replaced in DMEM media supplemented with 50 μg/ml gentamycin to ensure growth inhibition of extracellular bacteria. BMDM cell death levels were measured as a percentage of LDH release using the Promega CytoTox 96 Non-Radioactive Cytotoxicity Assay (G1780), according to manufacturer's instructions.

## Sublethal irradiation

Male and female $Mlkl^{S131P}$ colony mice used in this experiment were a mix of littermates and non-littermates aged 7–17 weeks, and wild-type mice that were littermates behaved equivalently to non-littermates. Mice were irradiated with a 5.5 Gy sub-lethal dose of γ-irradiation and received neomycin (2 mg/mL) in the drinking water for 3 weeks. At 4 days post irradiation, a mandible bleed was analyzed via ADVIA hematology to confirm successful irradiation. Mice had submandibular bleeds analyzed by ADVIA hematology and had long bones harvested for flow cytometry analysis at either 7, 14 or 21 days post-irradiation to assess stem cell capacity.

## Hematopoietic stem cell transplants

Donor bone marrow were injected intravenously into recipient female *C57BL/6-CD45*<sup>ly5.1</sup> mice (WEHI Kew) aged 6–10 weeks, following 11 Gy of γ-irradiation split over two equal doses. Recipient mice received neomycin (2 mg/mL) in the drinking water for 3 weeks. Long term capacity of stem cells was assessed by flow cytometric analysis of donor contribution to recipient mouse peripheral blood at 6 weeks.

## Zymosan-induced peritonitis

Male and female *Mlkl*<sup>S131P</sup> colony mice used in this experiment were a mix of littermates and non-littermates aged 7–17 weeks, and wild-type mice that were littermates behaved equivalently to non-littermates. 1 mg of zymosan A from *Saccharomyces cereviisiae* (Sigma-Aldrich) was intra-peritoneally injected into mice to induce sterile peritonitis. After 4 or 24 hours, mice were euthanized, cardiac bled and bone marrow collected. The peritoneal cavity was washed with 1 ml of cold PBS and cells within the lavage fluid were collected by centrifugation ($300 \times g$, 5 minutes). Bone marrow and peritoneum immune cells were quantified by flow cytometry.

## Flow cytometry

To analyze the innate and adaptive immune cells in peripheral blood, inguinal lymph nodes, spleen and bone marrow, isolated single cell suspensions were incubated with a combination of the following antibodies: CD4-BV421, CD8-PECy7, CD19-PerCPCy5.5, CD11b-BV510 or BV786, GR1-PE, CD45-Alexa700, Ly6G-V450 and Ly6C-APCCy7.

Splenic immune cells were analyzed at 14 day post *Salmonella* infection and incubated with a combination of the following antibodies: CD4-BV421, CD8-PeCy7, CD19-PerCPCy5.5, CD11b-BV510 or BV786, CD64-PE, CD45-Alexa700, Ly6G-V450 and Ly6C-APCCy7.

To analyze the contribution of donor and competitor cells in transplanted recipients, blood cells were incubated with either CD41-APC and Ter119-PE or Ly5.1-A700, Ly5.2-PE-Cy7, CD4-PE, CD8-PE, B220-BV650, Mac1-PerCPCy5.5. To analyze stem- and progenitor- cell compartment following sub-lethal irradiation, bone marrow cells were incubated with cKit-PerCPe710/PerCPCy5.5, CD150-A647, Sca1-APCCy7, B220-PE, CD19-PE, CD4-PE, CD8-PE, Gr1-PE. To analyze the relative percentages of stem and progenitor cells at steady state, bone marrow cells were incubated with cKit-PerCPCy5.5, Sca1-A594, CD150-BV421, CD105-PE, FcγRII-PECy7, and lineage markers (B220, CD19, CD4, CD8, GR1, Ly6G, Ter119)-A700. Finally, fluorogold (AAT Bioquest Cat#17514) was added for dead cell detection where appropriate. For detection of Annexin V at 21 days post-irradiation, bone marrow was incubated with Annexin V for 30 minutes.

Peritoneal and bone marrow immune cells at 4- or 24-hours post-zymosan injection were incubated with CD45-Alexa700, CD64-BV650, Ly6G-PE, Ly6C-APCCy7, F4/80-PerCPCy5.5, CD11b-BV510, CD8-PECy7, CD4-FITC, B220-APC and FC-blocker. Finally, propidium iodide (2 μg/mL, Sigma-Aldrich) was added for dead cell detection.

All samples were analyzed using the Aurora Cytex flow cytometer and FlowJo version 10.8.1. With exception of zymosan induced peritonitis and *Salmonella* experiments that used the Aurora Cytex automated volume calculator, cells were mixed with counting beads to quantify absolute cell numbers. All flow cytometry gating strategies are shown in Supplementary Fig. 7 and antibody details are outlined in Supplementary Table 1.

## Reactive oxygen species (ROS) detection

ROS was detected by mixing Chloromethyl-H$_2$DCFDA dye (1 μM; Invitrogen, #C6827) with bone marrow harvested from mice 21 days post-irradiation. Following a 30 minute incubation at 37 °C, loading buffer was removed and cells were placed into StemPro-34 plasma free medium (Thermofisher, #10639011) for 15 minute chase period. Cells were analysed using Aurora Cytex flow cytometer.

## Liposome dye release assays

Recombinant full-length human MLKL (residue 2–471) proteins were expressed in *Sf21* insect cells using bacmids prepared in DH10MultiBac *E.coli* (ATG Biosynthetics) from pFastBac Htb vectors using established procedures[70]. Briefly, full-length GST-tagged human MLKL proteins were purified using glutathione agarose resin (UBPBio)[45] followed by size-exclusion chromatography using HiLoad 16/600 Superdex 200 pg column (Cytivia). Fractions corresponding to full-length human MLKL tetramers (elution volume 55–63 ml) were pooled for liposome assays. Large Unilamellar Vesicles (LUV) 100 nm in diameter were prepared using a plasma membrane-like lipid mix (20% POPE, 40% POPC, 10% PI/PI(4,5)P$_2$, 20% POPS, 10% POPG) resuspended in chloroform at 20 mg/mL. To form dye filled liposomes, dried lipids were resuspended in LUV buffer (10 mM HEPES pH 7.5, 135 mM KCl) supplemented with 50 mM 5 (6)- Carboxyfluorescein dye (Sigma). Freeze-thaw cycles were completed five times prior to extrusion (21 times) of the lipid mixture through polycarbonate membranes of 100 nm size cutoff (Avanti Polar Lipids, AL USA) using a pre-warmed mini extruder (Avanti Polar Lipids, AL USA). Liposome stocks were stored at 4 °C in the dark. Recombinant human MLKL protein was diluted to 1 μM (2× desired final concentration) in LUV buffer and aliquoted into a 96-well flat-bottom plate (ThermoFisher Scientific). Liposomes were purified from excess dye using a PD-10 desalting column (Cytiva) and diluted to 20 μM in LUV buffer. Immediately following addition of the liposomes to the plate (1:1 ratio liposomes:protein) fluorescence (485 nm excitation, 535 nm emission) was measured every 2 minutes for 60 minutes total on the CLARIOstar plate reader (BMG Labtech; 5.70 R2). Baseline measurements were determined by incubation of liposomes with LUV buffer alone. All assays were performed in triplicate. Data plotted as mean ± SD of one independent repeat that is representative of three independent assays.

## Statistical analyses

All data were graphed using Prism v9. Individual points plotted signify independent experimental repeats performed on different days or biologically independent data points. All *P* values were calculated in Prism v9 using the statistical test identified in figure legends.

## Reporting summary

Further information on research design is available in the Nature Portfolio Reporting Summary linked to this article.

## Data availability

The biological materials generated during this study are available from the corresponding authors upon reasonable request. Experimental data generated in this study are provided in the Supplementary Information and Source Data file. Relevant human sequence data can be directly accessed from the Sequence Read Archive (SRA, BioProject accession number PRJNA1007397) via the following link; https://www.ncbi.nlm.nih.gov/sra/?term=PRJNA1007397. Publicly available gnomAD data used can be found at https://gnomad.broadinstitute.org/gene/ENSG00000168404?dataset=gnomad_r2_1. Publicly available PDB data can be found at; 2MSV [https://doi.org/10.1016/j.str.2014.07.014], 4MWI [https://doi.org/10.1042/BJ20131270] and 4BTF [https://doi.org/10.1016/j.immuni.2013.06.018]). Source data are provided with this paper.

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

## Acknowledgements

We thank all the following people for their technical assistance; Aira Nuguid and Tina Cardamone (Phenomics Australia Histopathology and Slide Scanning Service- The University of Melbourne). WEHI Cytometry Facility, WEHI Antibody Facility, WEHI Centre for Dynamic Imaging, WEHI Bioservices, Cheree Fitzgibbon (WEHI), Jacinta Hansen (WEHI) Jingjing Vivian Tan and Yafei Zhang (ANU, The Australian Genomics Health Alliance). The generation of $Mlkl^{S131P}$ mice by CRISPR/Cas9 gene editing was performed by Andrew Kueh and Marco Herold (WEHI MAGEC laboratory) supported by the Australian Phenomics Network (APN) and the Australian Government through the National Collaborative Research Infrastructure Strategy (NCRIS) program. We thank Warren Alexander, Mary Speir, and Melanie Bahlo for the provision of important resources and expertize. We thank Michael Hildebrand and Tom Witkowski from Epilepsy Research Centre, Department of Medicine, Austin Health for assistance with Sanger sequencing. We are grateful to the National Health and Medical Research Council for fellowship (J.M.H., 1142669; A.L.S., 2002965; J.M.M., 1172929; J.S., 1107149), grant (J.M.M., 1105023; K.R.M., 1092602; J.S., 1105023; J.M.H., 2011584) and infrastructure (IRIISS 9000719); Arthritis Australia support to K.R.M; K.E.L funding by Future Fellowships from the ARC (FT19010266). We acknowledge scholarship support for S.E.G (Australian Government Research Training Program Stipend Scholarship; Wendy Dowsett Scholarship), Y.M (Melbourne Research Scholarship; AINSE PGRA Scholarship), D.F (Australian Government Research Training Program Stipend Scholarship), A.V.J (Australian Government Research Training Program Stipend Scholarship), and S.C (Walter and Eliza Hall Handman PhD Scholarship). Victorian State Government Operational Infrastructure Support Scheme.

## Author contributions

Conceptualization: S.E.G., J.S., J.M.M., J.M.H. Methodology: S.E.G., K.R.M., M.K., V.J., R.A., V.E., S.C., Y.M., D.F., E.C.T.C, K.M.P., A.V.J., G.K.A., L.D., M.D., C.R.H., C.H., S.N.Y., K.E.L., I.P.W., G.E., A.P.N., J.S.P., A.L.S., J.M.H. Resources: K.R.M., M.K., M.C., V.A., C.V., C.A.S., J.S.P., J.S., J.M.M., J.M.H. Supervision: K.M., M.K., J.S., J.M.M., J.M.H. Funding acquisition: J.S., J.M.M., J.M.H. S.E.G and J.M.H co-wrote the paper with input from authors.

## Competing interests

S.E.G, K.M.P, A.L.S, C.R.H, S.N.Y, J.S, J.M.M, and J.M.H contribute or have contributed, to a project developing necroptosis inhibitors in collaboration with Anaxis Pty Ltd. K.R.M received funding from CSL Pty Ltd. The remaining authors declare no competing interests.

## Additional information

[1]The Walter and Eliza Hall Institute, Parkville, VIC, Australia. [2]University of Melbourne, Department of Medical Biology, Parkville, VIC, Australia. [3]Centre for Innate Immunity and Infectious Diseases, Hudson Institute of Medical Research, Clayton, VIC, Australia. [4]Department of Molecular and Translational Science, Monash University, Clayton, VIC, Australia. [5]Department of Microbiology, Monash University, Clayton, VIC, Australia. [6]University of Melbourne, Faculty of Medicine, Dentistry and Health Sciences, Parkville, VIC, Australia. [7]Centre for Personalised Immunology and Canberra Clinical Genomics, Australian National University, Canberra, ACT, Australia. [8]Department of Immunology and Infection, John Curtin School of Medical Research, Australian National University, Canberra, ACT, Australia. [9]Institute of Virology, Technical University of Munich/Helmholtz Munich, Munich, Germany. [10]Clinical Haematology Department, The Royal Melbourne Hospital and Peter MacCallum Cancer Centre, Parkville, VIC, Australia. [11]Department of Clinical Immunology & Allergy, Royal Melbourne Hospital, Parkville, VIC, Australia. [12]Present address: Cambridge Institute for Therapeutic Immunology and Infectious Disease, University of Cambridge, Cambridge, UK. [13]Present address: The Francis Crick Institute, London, UK. [14]Present address: University College London, London, UK. [15]Present address: China Australia Centre for Personalized Immunology (CACPI), Renji Hospital, Shanghai Jiao Tong University School of Medicine (SJTUSM), Shanghai, China. ✉e-mail: jhildebrand@wehi.edu.au

