## [Peer Review File · Nature Communications]

A common human MLKL polymorphism confers resistance to negative regulation by phosphorylationREVIEWER COMMENTS

Reviewer #1 (Remarks to the Author):

This manuscript by Garnish et al, is an interesting and compelling study on a single mutation in MLKL that has been identified in humans. The authors demonstrate how this mutation confers resistance to regulation of MLKL and also investigated its role in disease and injury using a newly developed mouse model. As MLKL is fundamental to the inflammatory cell death pathway of necroptosis, and gene variants of MLKL have been observed in humans, this study is timely and informative. The strengths of the study include thorough analysis and use of appropriate controls and tools to investigate how the Mlkl S132P/S131P polymorphism regulates cell death kinetics in vitro and in vivo. The authors shed light on how this mutation renders cells somewhat more resistant to standard methods of inducing necroptosis, using TNF, but more sensitive to IFN β , perhaps via its ability to induce expression of Mlkl. The work is well described and the manuscript well written and easy to follow and appreciate for the amount of work put into the project and the rigor of the studies. Several questions may be important and interesting for the authors to comment on or address:

1. Is it known if individuals carry multiple gene variants that could potentially result in a fairly normal phenotype (as what was seen in Figure 1 with cells carrying both the S83D mutation and the S132P mutation)?
2. When reading the results, a central question that came to mind was why MLKL was expressed at lower levels in the strain carrying the mutant. This was addressed by the authors in the Discussion, but an additional explanation may be reduced type I IFNs or tonic signaling in these cells. Were there any differences in the cytokine profiles present in the three strains (at steady-state or upon infection with salmonella or in response to radiation or zymosan-induced peritonitis)?
3. The authors suggest an intriguing possibility wherein heterozygotes may have an advantage to immunologic stress, which would argue why the mutation is present in the population at all (especially where, at least in the example of salmonella, the homozygous mutants have reduced clearance ability). The most notable place where a "heterozygote advantage" was observed was day 7 post-radiation, and it was specifically the observation that hets exhibited increased blood monocytes (panel C). This was one time point where fewer mice were included and should this be a significant increase (as would be expected if more animals were used), the authors could make a compelling argument in favor of carriers of this mutation having a unique and specific response to hematopoietic injury.
4. The observation of reduced monocytes in the S131P homozygous mutants is intriguing as well. Did the authors determine if there was a reduction in cMoPs (common monocyte progenitors)? Is there an intrinsic bias to the HSCs/HSPCs that may favor granulocytes over monocytes? Performing in vitro CFU assays (methocult) for myeloid progenitors may help address this. Do the authors have any insight as to why monopoiesis may be specifically impacted? To make a claim that monopoiesis is impacted, additional experiments are necessary. It is also important to note that turnover of intestinal macrophages requires circulating, HSC-derived monocytes, and since diet/microbiome is so critical in regulating hematopoiesis, it may be worth investigating monocytes in the intestine of these mutants.
5. Additional questions about the impact of the mutation on monocytes includes questions about the peritonitis model (Figure 4). Though the authors conclude that reduced monocytes in the peritonitis experiments are likely due to reduce monocytes in the bone marrow, this is not completely supported by the data. Monocytes in the peripheral blood were similar in all groups and actually higher in the homozygous mutants at the early time point (Panel E), suggesting that recruitment to the site of inflammation was reduced. Were there differences in chemokines and attractants that would impact recruitment of monocytes (or neutrophils) to the peritoneum? (This last comment is related to point 2 above.) Do peritoneal macs exist at similar frequencies in the mutant mice??

Minor:

1. Flow data in supplement should be labeled more clearly (in addition to fluor, include the marker being studied). Plots are difficult to see with the gate names frequencies placed on top of the plots.

2. There were some noticeable outliers in the data where 2 mice were very different from the group. In these cases, were there sex-dependent variations? Were there correlations with other observations? For example, in Figure 4, were reduced neutrophils at 24 hours (peritoneum in the hets) observed in the same mice that had elevated blood neutrophils at this time point (panels C and E; these data again beg questions about expression of key cytokines and chemokines). Another example: In Figure 5: extreme outliers for numbers of B cells or monocytes, did these mice have higher/lower bacterial burdens?? Were mice consistent across tissues with bacterial burden or did some mice exhibit high splenic burden (1 was much higher than all others) but lower burden in other tissues? There appears to be somewhat of a bias in the data due to 1 or 2 data points in the spleen liver and feces with very high burdens and it would be important to note if this was the same mouse in all cases.

Reviewer #2 (Remarks to the Author):

In this manuscript, Garnish and colleagues characterised a new human MLKL variant, carrying a Serine132 to Proline substitution mutation. This variant is present in about 2 - 3 % of the population. The authors identified two MLKLS132P carriers that suffer distinct inflammatory/autoimmune disorders. They therefore suggest that this MLKLS132P polymorphism might be the underlying cause of their condition. Consistent with this hypothesis, they find that overexpression of MLKLS132P renders human cells susceptible to necroptosis. They then generate a knock-in mouse line to characterise the physiological role of this polymorphism. While MLKLS132P mutant mice do not show any gross abnormalities and are born at expected ratios, they find that cells from this mutant MLKLS132P line are more resistant to necroptosis *in vitro*. Interestingly, MLKLS132P mutant mice have a lower population of Ly6Chigh monocytes in their bone marrow. By employing models of radiation-induced cell depletion, Salmonella infection and zymosan-triggered peritonitis, the authors note key differences in the immune cell composition of the MLKLS132P mice compared to WT, though many of the phenotypes described seem to be reminiscent from the alterations found at steady state.

Overall, this is an interesting manuscript that describes a new MLKL polymorphism. However, I find that it is somewhat preliminary as many key issues remain unclear. For example 1) how does the S132P substitution mutation contribute to the disease pathology in human and mouse; 2) why does the S132P mutation sensitises cells to necroptosis in humans while it seems to be protective in mice; and 3) what is the molecular mechanism for the phenotypes seen *in vivo* in the MLKLS132P animals.

Major comments:

1. The link between the MLKLS132P polymorphism and human disease is very weak. Given that this polymorphism is relatively common in humans (2-3 %), the characterisation of 2 individuals does not seem sufficient to establish a link. The authors need to expand their analysis in order to support their claims. The two patients described in the manuscript have very different symptoms, with the only common feature between the two being that these are diseases associated with inflammation. Considering the prevalence of this polymorphism proposed by the authors of 2-3% of the population, it seems unlikely that this is the causative mutation of the observed phenotypes. Regardless, the authors would have to provide more patient data to make that claim. The current *in vitro* data presented on cytokine production by PBMCs has an n=1, and the plot suggests only a marginal difference in cytokine production by the MLKLS132P-carrier, which would unlikely explain the inflammatory phenotype seen in the patient (Supp. Fig 1A, B). The authors need to expand their cohort of both carriers of the MLKLS132P polymorphism and controls to support their claims that the MLKLS132P is linked to a higher cytokine production in PBMCs. Alternatively, the authors might want to focus on the mouse model only.

2. The authors evaluate the consequence of the MLKLS132P mutation in mice using three different experimental settings. However, this mutation doesn't seem to induce a clear alteration in

responses in vivo, and the data is currently presented in a disconnected way, with not a lot of in-depth insight into the mechanisms of the phenotypes observed. Below are a few comments:

a. What is the explanation for the immune alterations seen in MLKLS132P animals at steady state? Is this related to cell death induced by MLKL or are there other signalling pathways downstream of MLKL that could explain the phenotype, such as vesicle shedding? Could this have a role in the immune alterations observed?

b. What causes the lack of competitiveness of MLKLS132P cells? This is somewhat surprising considering that MLKLS132P cells should be less prone to death in vitro? What is the role of RIPK3 in this phenotype?

c. The phenotypes caused by exposure to zymosan and salmonella seem to reflect the differences in immune cells already observed at steady state, particularly with respect to the monocyte population. How much does this dataset add to the manuscript, to justify each of them being a full main figure?

3. The authors describe that the human MLKLS132P protein is less prone to inhibition by the MLKL inhibitor, NSA. Is this also the case for the MLKL inhibitor BI-8925, which targets Cys86 (instead of Cys88 by NSA)?

4. Fig 1D, 1H and Suppl Fig 1E, F: the authors should analyse longer time points? It seems from their graph that MLKLWT expressing cells are also starting to die in the presence of NSA. This could suggest that the "breakthrough cell death" is not specific to the MLKLS132P mutant, but instead associated with the inhibitor losing its potency (being degraded).

5. Why does the MLKLS132P mutation enhances necroptosis in human cells but suppresses necroptosis in mouse cells? This is a key point that needs to be addressed. A key difference between the two systems is that the authors over-express MLKL in human cells, while the murine setting is based on a knock-in system. Hence, they are not really comparable. More data is needed here to corroborate that the human system behaves differently (e.g. more cell types, primary cells instead of cancer cells).

6. Fig 2G suggest that IFN β /TNF treatment induces RIPK1-independent cell death in MLKLS132P cells. How do these cells die? What is the involvement of RIPK3?

7. The Incucyte data shown by the authors in the mouse cells seem to represent a quantification of PI+ cells per area. This is not appropriate to compare the different genotypes used in their experiment as it does not correct for different cell numbers in their wells. The authors should analyse this data as percentage of total cells in the well, like they do in the panels with human cells. Furthermore, across all Incucyte data, the authors should include statistics on their plots to validate the differences observed.

We thank the reviewers for their thoughtful, constructive and detailed feedback on our manuscript. In response, we have added to new data in **Figures 1H, 2G, 2H, 3G, 3H, 4G, 5F, 5G** and **Supplementary figures 1M, 1N, 3C-G & 4W-Y**. We have also edited the main text in responds to reviewer comments (as detailed below); these changes are **highlighted in yellow** in the revised manuscript. In our point-by-point response to reviewers below, the reviewers' remarks are presented verbatim in *blue italics* and our response to reviewers' comments are shown in black text.

Reviewer #1 (Remarks to the Author):

This manuscript by Garnish et al, is an interesting and compelling study on a single mutation in MLKL that has been identified in humans. The authors demonstrate how this mutation confers resistance to regulation of MLKL and also investigated its role in disease and injury using a newly developed mouse model. As MLKL is fundamental to the inflammatory cell death pathway of necroptosis, and gene variants of MLKL have been observed in humans, this study is timely and informative. The strengths of the study include thorough analysis and use of appropriate controls and tools to investigate how the Mkl1 S132P/S131P polymorphism regulates cell death kinetics in vitro and in vivo. The authors shed light on how this mutation renders cells somewhat more resistant to standard methods of inducing necroptosis, using TNF, but more sensitive to IFN β , perhaps via its ability to induce expression of Mkl1. The work is well described and the manuscript well written and easy to follow and appreciate for the amount of work put into the project and the rigor of the studies. Several questions may be important and interesting for the authors to comment on or address;

1. Is it known if individuals carry multiple gene variants that could potentially result in a fairly normal phenotype (as what was seen in Figure 1 with cells carrying both the S83D mutation and the S132P mutation)?

This is a very interesting possibility. According to the largest available repositories of human genomic variant data, gnomAD and the UK Biobank, no individuals have been recorded with a polymorphism that encodes the p.Ser83Asp (S83D) replacement. There are however individuals that carry the following closely related changes:

p.Ser83Cys (MAF 3.54e10⁻⁵)

p.Arg82Ser (MAF 1.57e10⁻³)

While the likelihood of an individual carrying one of these gene variants in combination with p.Ser132Pro is low (assuming random mating within a population, and with the caveat that published 'global' frequencies do not reflect the actual ancestry distribution of the human population), we have modelled the co-occurrence of these natural human gene variants through compound mutation and inducible expression in *MLKL*^{-/-} HT29 cells. Now included as **Supplementary Figures 1M & N**, p.Ser83Cys is functional, whilst p.Arg82Ser is resistant to necroptotic cell death, phenocopying the rationally designed p.Ser83Asp (S83D) replacement. Consistent with our S83D findings, combining the p.Arg82Ser variant with p.Ser132Pro leads to restoration of necroptotic killing capacity.

2. When reading the results, a central question that came to mind was why MLKL was expressed at lower levels in the strain carrying the mutant. This was addressed by the authors in the Discussion, but an additional explanation may be reduced type I IFNs or tonic signaling in these cells. Were there any differences in the cytokine profiles present in the three strains (at steady-state or upon infection with salmonella or in response to radiation or zymosan-induced peritonitis)?

We thank the reviewer for prompting further investigation on this point. To address the potential for different basal cytokine levels and signaling as a potential explanation for reduced levels of MLKL^{S131P} in cells derived from the *MLK^{S131P}* mouse strain, we performed multiplex analysis for 23 mouse cytokines on serum and peritoneal lavage fluid. These data are now included in **Figure 2H**, **Figure 3G**, **Supplementary Figure 3E**, **Figure 4G**, **Supplementary Figure 4W-Y** and **Figure 5F,G**. At basal state we observed no differences between wild-type, heterozygote, or homozygote mice in serum cytokines (**Figure 2H**). However, following myelosuppressive radiation, zymosan-induced peritonitis, and infection with *Salmonella* we observed the following variations in cytokine profiles:

- increased MCP-1 levels following zymosan-induced peritonitis (peritoneal lavage, **Figure 4G**) and *Salmonella* infection (serum, **Figure 5F**)

- increased serum G-CSF at 14- and 21-days post-irradiation (**Figure 3G**).

These differences in *MLK^{S131P/S131P}* cytokine profile support our major findings in **Figures 3-5**, although these observations do not provide an alternative explanation for the reduced *MLKL* expression level.

3. The authors suggest an intriguing possibility wherein heterozygotes may have an advantage to immunologic stress, which would argue why the mutation is present in the population at all (especially where, at least in the example of salmonella, the homozygous mutants have reduced clearance ability). The most notable place where a “heterozygote advantage” was observed was day 7 post-radiation, and it was specifically the observation that hets exhibited increased blood monocytes (panel C). This was one time point where fewer mice were included and should this be a significant increase (as would be expected if more animals were used), the authors could make a compelling argument in favor of carriers of this mutation having a unique and specific response to hematopoietic injury.

We agree that the presence of a hematopoietic advantage for heterozygotes is a compelling argument for the accumulation of this polymorphism in humans. Due to limited mouse numbers, we were not able to add additional numbers of homozygotes to the sub-lethal irradiation experiment however, we were able to add additional wild-type and heterozygote mice to the day 7 timepoint. With the additional numbers in **Figure 3C** the numbers of circulating monocytes at 7 days post-irradiation are now more comparable between wild-type and heterozygote.

We also performed additional competitive bone marrow transplants with *Mikl*^{WT/S131P} stem cells alongside wild-type and homozygote controls to test for a heterozygote advantage. These data are now added to **Figure 3H, Supplementary Figure 3F,G**. In this context, *Mikl*^{WT/S131P} stem cells do trend towards giving rise to a higher percentage of red blood cells and platelets. The strongest evidence for a heterozygote advantage over WT counterparts remains for total nucleated cell and LSK cell numbers 21 days following sub-lethal irradiation (**Figure 3D, F**).

4. The observation of reduced monocytes in the S131P homozygous mutants is intriguing as well. Did the authors determine if there was a reduction in cMoPs (common monocyte progenitors)?

We thank the reviewer for prompting further study of the Ly6C^{hi} monocyte defect in the bone marrow of homozygote mice at steady state. We examined the relative percentage of granulocyte-macrophage progenitors (GMP; Lin⁻cKit⁺Sca1⁻CD150⁻FcyRII/II⁺), megakaryocyte-erythroid progenitors (MegE; Lin⁻cKit⁺Sca1⁻CD150⁺FcyRII/II⁻CD105⁻), colony-forming units-erythroid progenitors (CFU-E; Lin⁻cKit⁺Sca1⁻CD150⁻FcyRII/II⁻CD105⁺) and pre-granulocyte-macrophage progenitors (pre-GM; Lin⁻cKit⁺Sca1⁻CD150⁻FcyRII/II⁻CD105⁻) in the bone marrow of wild-type and homozygote mice. No significant differences in these progenitor populations were observed between genotypes. These data have been added to **Supplementary Figure 2** in panels **T & U** and are addressed on lines 217-218 of the results section.

Is there an intrinsic bias to the HSCs/HSPCs that may favor granulocytes over monocytes? Performing in vitro CFU assays (methocult) for myeloid progenitors may help address this.

This is an interesting question. We performed CFU assays examining the capacity of HSCs to differentiate and establish colonies under four different stimuli: GCSF, GMCSF, MCSF and a combination of SCF, IL-3 and EPO (SIE). These CFU assays were blind-scored for the formation of blast, eosinophil, granulocyte, granulocyte-macrophage, macrophage, and megakaryocyte colonies. No intrinsic bias favoring granulocytes over monocytes was observed for *Mikl*^{S131P/S131P} HSCs. However, we did observe increased macrophage colony formation from *Mikl*^{S131P/S131P} HSCs, when compared with *Mikl*^{WT/WT}, under SIE conditions. These data are now included in **Figure 2G** and are addressed on lines 212-220 of the results section.

Do the authors have any insight as to why monopoiesis may be specifically impacted? To make a claim that monopoiesis is impacted, additional experiments are necessary. It is also important to note that turnover of intestinal macrophages requires circulating, HSC-derived monocytes, and since diet/microbiome is so critical in regulating hematopoiesis, it may be worth investigating monocytes in the intestine of these mutants.

We thank the reviewer for this suggestion. To address this, we harvested ileum samples from unchallenged wild-type and homozygote mice and quantified the number of F4/80 positive cells via image analysis following immunohistochemical staining. Representative images and quantification are presented below in **Response Figure 1**. We observed no alterations in the average number of F4/80 positive cells between *Mikl* wild-type and *Mikl*^{S131P/S131P} homozygotes in the sections examined.

Response Figure 1: (A) Representative H&E- and F4/80-stained sections of ileum from *Mik1*^{WT/WT} and *Mik1*^{S131P/S131P} mice at steady state. Sections are representative of $n = 8$ mice per genotype (4 male, 4 female). **(B)** Quantification of F4/80⁺ cells in ileum sections presented as % of total cells or raw count. Data is presented as mean of $n = 8$.

Overall, whilst these three independent lines of enquiry do not directly identify the cause of reduced Ly6C^{hi} monocyte populations in the bone marrow, it leads us to propose that the cause is not defective monopoiesis. As such we have removed reference to ‘defective monopoiesis’ and propose that differences in monocyte dispersal and differentiation outside the bone marrow is an important future line of investigation (lines 382-384 of the discussion).

5. Additional questions about the impact of the mutation on monocytes includes questions about the peritonitis model (Figure 4). Though the authors conclude that reduced monocytes in the peritonitis experiments are likely due to reduce monocytes in the bone marrow, this is not completely supported by the data.

We appreciate the reviewer bringing this to our attention, and completely agree with their interpretation. As such we have reworded our analysis of the reduced monocytes in the peritonitis model on lines 297-303.

Monocytes in the peripheral blood were similar in all groups and actually higher in the homozygous mutants at the early time point (Panel E), suggesting that recruitment to the site of inflammation was reduced. Were there differences in chemokines and attractants that would impact recruitment of monocytes (or neutrophils) to the peritoneum? (This last comment is related to point 2 above.) Do peritoneal macs exist at similar frequencies in the mutant mice??

Again, we thank the reviewer for their experimental suggestion and have tested for the abundance of inflammatory chemokines in the peritoneum following zymosan injection. We also provide the below data (**Response Figure 2**) to show that peritoneal macrophages exist at similar frequencies between $Mikl^{WT/WT}$ and $Mikl^{S131P/S131P}$ mice at steady state.

Response Figure 2. Peritoneal macrophages exist at similar frequencies in $Mikl^{WT/WT}$ and $Mikl^{S131P/S131P}$ mice at steady state. Flow cytometry of $CD64^+$ and $F4/80^+$ peritoneal macrophages at steady state. Each symbol represents one independent animal. Error bars represent mean \pm SEM for $n = 3 - 7$ mice as indicated.

Minor:

1. Flow data in supplement should be labeled more clearly (in addition to fluor, include the marker being studied). Plots are difficult to see with the gate names frequencies placed on top of the plots.

We thank the reviewer for their suggestion. The flow data in the supplement have been re-labeled providing both fluorophore and marker being studied, with gate names now placed besides rather than on top of the plots.

2. There were some noticeable outliers in the data where 2 mice were very different from the group. In these cases, were there sex-dependent variations? Were there correlations with other observations? For example, in Figure 4, were reduced neutrophils at 24 hours (peritoneum in the hets) observed in the same mice that had elevated blood neutrophils at this time point (panels C

and E; these data again beg questions about expression of key cytokines and chemokines). Another example: In Figure 5: extreme outliers for numbers of B cells or monocytes, did these mice have higher/lower bacterial burdens?? Were mice consistent across tissues with bacterial burden or did some mice exhibit high splenic burden (1 was much higher than all others) but lower burden in other tissues? There appears to be somewhat of a bias in the data due to 1 or 2 data points in the spleen liver and feces with very high burdens and it would be important to note if this was the same mouse in all cases.

The reviewer has raised an interesting point and we have now analyzed outlier datapoints and how they correlate with other findings within the experiment. Where there were multiple outliers within one genotype in datasets throughout figures 2-5 these did not correlate with sex-dependent variations. This is highlighted in the sex-stratified data presented in accompanying supplementary figures 2,4 and 5. In response to the reviewers' examples:

- Figure 4: The heterozygous mice with the lowest numbers of peritoneal neutrophils at 24 hours are the same mice that had elevated blood neutrophils at this timepoint.
- Figure 5: Whilst some of these outliers for the B cells or monocyte numbers do have high bacterial burdens, there is no consistent correlation between increased B or monocyte numbers and tissue bacterial burden.
- Figure 5: Bacterial burdens were consistent across tissues, mice exhibiting high burdens in one organ did not exhibit the lowest burdens in another. For example: The homozygote mouse with the highest bacterial burden in the spleen, has a bacterial burden above the mean in the liver. Furthermore, the highest burdens values observed in spleen, liver and feces (Figure 5B) do not correlate to a single mouse.

Reviewer #2 (Remarks to the Author):

In this manuscript, Garnish and colleagues characterised a new human MLKL variant, carrying a Serine132 to Proline substitution mutation. This variant is present in about 2 - 3 % of the population. The authors identified two MLKLS132P carriers that suffer distinct inflammatory/autoimmune disorders. They therefore suggest that this MLKLS132P polymorphism might be the underlying cause of their condition. Consistent with this hypothesis, they find that overexpression of MLKLS132P renders human cells susceptible to necroptosis. They then generate a knock-in mouse line to characterise the physiological role of this polymorphism. While MLKLS132P mutant mice do not show any gross abnormalities and are born at expected ratios, they find that cells from this mutant MLKLS132P line are more resistant to necroptosis in vitro. Interestingly, MLKLS132P mutant mice have a lower population of Ly6Chigh monocytes in their bone marrow. By employing models of radiation-induced cell depletion, Salmonella infection and zymosan-triggered peritonitis, the authors note key differences in the immune cell composition of the MLKLS132P mice compared to WT, though many of the phenotypes described seem to be reminiscent from the alterations found at steady state.

Overall, this is an interesting manuscript that describes a new MLKL polymorphism. However, I find that it is somewhat preliminary as many key issues remain unclear. For example 1) how does

the S132P substitution mutation contribute to the disease pathology in human and mouse; 2) why does the S132P mutation sensitises cells to necroptosis in humans while it seems to be protective in mice; and 3) what is the molecular mechanism for the phenotypes seen in vivo in the MLKLS132P animals.

Major comments

1. The link between the MLKLS132P polymorphism and human disease is very weak. Given that this polymorphism is relatively common in humans (2-3 %), the characterisation of 2 individuals does not seem sufficient to establish a link. The authors need to expand their analysis in order to support their claims. The two patients described in the manuscript have very different symptoms, with the only common feature between the two being that these are diseases associated with inflammation. Considering the prevalence of this polymorphism proposed by the authors of 2-3% of the population, it seems unlikely that this is the causative mutation of the observed phenotypes. Regardless, the authors would have to provide more patient data to make that claim. The current in vitro data presented on cytokine production by PBMCs has an n=1, and the plot suggests only a marginal difference in cytokine production by the MLKLS132P-carrier, which would unlikely explain the inflammatory phenotype seen in the patient (Supp. Fig 1A, B). The authors need to expand their cohort of both carriers of the MLKLS132P polymorphism and controls to support their claims that the MLKLS132P is linked to a higher cytokine production in PBMCs. Alternatively, the authors might want to focus on the mouse model only.

We completely agree with the reviewer's comments regarding the need for additional independent patient data. Indeed, we anticipate that our findings will prompt further investigations of MLKL polymorphisms and their potential links to disease, which should identify further patients for future studies. While we were careful to note that the evidence for a potential pathogenic role for this polymorphism in humans is still limited and difficult to demonstrate given its high frequency, we have moved all patient data to **Supplementary Figure 1** to avoid over-interpretation of these patient findings by readers.

2. The authors evaluate the consequence of the MLKLS132P mutation in mice using three different experimental settings. However, this mutation doesn't seem to induce a clear alteration in responses in vivo, and the data is currently presented in a disconnected way, with not a lot of in-depth insight into the mechanisms of the phenotypes observed. Below are a few comments:
a. What is the explanation for the immune alterations seen in MLKLS132P animals at steady state? Is this related to cell death induced by MLKL or are there other signalling pathways downstream of MLKL that could explain the phenotype, such as vesicle shedding? Could this have a role in the immune alterations observed?

The reviewer has raised an important consideration and in response to both reviewers we have completed several independent experiments to dissect the underlying cause of the *Mkl^{S131P/S131P}* immune cell alterations at steady state. We used multiplex assays to investigate cytokine profiles in our mice at steady state and these data were added to **Figure 2H**. We also examined the GM

progenitor populations in our mice at steady state and these data were added to **Supplementary Figure 2T, U**. While neither of these yielded any overt differences in circulating cytokine levels or myeloid progenitor populations at steady state, colony forming assays did reveal an important difference that may contribute to immune alterations observed in *Mikl*^{S131P/S131P} mice. *Mikl*^{S131P/S131P} bone marrow HSCs formed increased numbers of macrophage-like colonies under SCF, IL-3 and EPO (SIE) stimulation. Overall, whilst these three independent lines of enquiry do not directly identify the cause of reduced Ly6C^{hi} monocyte populations in the bone marrow, we propose that differences in monocyte dispersal and differentiation outside the bone marrow is an important future line of investigation (lines 382-384 of the discussion).

b. What causes the lack of competitiveness of MLKLS132P cells? This is somewhat surprising considering that MLKLS132P cells should be less prone to death in vitro? What is the role of RIPK3 in this phenotype?

We thank the reviewer for prompting clarification of this important point. We have added data to **Supplementary Figure 3 (C, D)** that demonstrate increased ROS and Annexin V in *Mikl*^{S131P/S131P} stem cells at 21-days post-irradiation when compared to *Mikl*^{Wt/Wt} littermate controls. This phenotype is consistent with our previous reports for the constitutively active *Mikl*^{D139V} mutant (Hildebrand *et al.*, 2020 *Nat. Comm.*). We posit that in the highly inflammatory environment that results from myelosuppressive irradiation, an increase in the presence of cytokines like IFN- β enhances *Mikl* gene expression and protein production. Constitutively active forms of MLKL, which can be turned over before causing cell death below a threshold, eventually overwhelms cellular neutralisation/clearance mechanisms to cause cell death. This phenomenon also explains the gene-dose dependent, selective toxicity of IFN- β in **Figure 2I** for MDFs isolated from *Mikl*^{S131P} mice.

c. The phenotypes caused by exposure to zymosan and salmonella seem to reflect the differences in immune cells already observed at steady state, particularly with respect to the monocyte population. How much does this dataset add to the manuscript, to justify each of them being a full main figure?

We agree with the reviewer that these models do reflect the steady state differences observed. It is however important to demonstrate clearly to readers, within the main body of the manuscript, that the capacity for short term immune cell recruitment and resolution was examined in this study. Likewise, given the high frequency of this *MLKL* polymorphism in the human population, the role such a mutation may play in a more 'real world' infection scenario (i.e, oral route bacterial infection) is a common question we receive and thus of broad interest to prospective readers.

Importantly, *Mikl* is a highly inducible gene capable of killing cells when induced above a threshold (e.g. Rodriguez *et al.* *CDD* 2015, Sarhan *et al.* *CDD* 2018, Hildebrand *et al.* *Nat. Comm.* 2020). A reflection of the steady state phenotype is thus not necessarily a predictable outcome in these challenge models, further supporting their placement within the main manuscript.

3. The authors describe that the human MLKLS132P protein is less prone to inhibition by the MLKL inhibitor, NSA. Is this also the case for the MLKL inhibitor BI-8925, which targets Cys86 (instead of Cys88 by NSA)?

As requested by the reviewer, we examined the inhibition of $MLKL^{WT}$ and $MLKL^{S132P}$ with inhibitor BI-8925. In the below figure (Response Fig. 3), BI-8925 is not as potent as NSA in HT29 cells. Breakthrough death was observed in HT29s expressing both $MLKL^{WT}$ and $MLKL^{S132P}$ however HT29s expressing $MLKL^{S132P}$ exhibited increased maximal death and kinetics. This is consistent with our NSA findings. We have not included these data in the manuscript itself, but these data will be available to readers through the publication of the reviews and responses with the paper.

Response figure 3: $MLKL^{S132P}$ exhibits increased cell death under BI-8925 inhibition. $MLKL^{-/-}$ HT29 cells expressing doxycycline (dox) inducible $MLKL^{WT}$ or $MLKL^{S132P}$ were treated with dox; 100 ng/ml and necroptotic stimulus (TNF, Smac mimetic, IDN-6556; TSI) in the presence of $MLKL$ inhibitor BI-8925 (1 or 10 μ M). Cell death was measured at hour intervals for 21 hours by percentage of SYTOX-green positive cells using IncuCyte SX5 live cell imaging. Independent cell lines were assayed in n=4 experiments, with error bars indicating the mean \pm SEM.

4. Fig 1D, 1H and Suppl Fig 1E, F: the authors should analyse longer time points? It seems from their graph that $MLKL^{WT}$ expressing cells are also starting to die in the presence of NSA. This could suggest that the “breakthrough cell death” is not specific to the $MLKL^{S132P}$ mutant, but instead associated with the inhibitor losing its potency (being degraded).

We agree that it would be ideal to increase the length of analysis time for experiments presented in Figures 1D, H and Suppl 1E,F. In our experience, dead HT29 cells eventually form clumps and lift off the surface of the plate (**Response Figure 4A**). The IncuCyte imager is no longer able to reliably detect these, making calculations of % dead cells less accurate after 24 hrs.

To address the reviewer’s valid concerns regarding the stability (thus potency) of NSA, we show here an experiment where we use tenfold the amount of NSA, treating cells both at time zero and again at 8hr (**Response Figure 4B**). Under these conditions, where no breakthrough necroptosis is evident for cells expressing $MLKL^{WT}$, cells expressing $MLKL^{S132P}$ retain the capacity to overcome NSA inhibition.

Response figure 4: MLKL^{S132P} exhibits breakthrough death in the presence of additional NSA treatment. (A) Phase image of *MLKL*^{-/-} HT29 cells expressing MLKL^{S132P} at 24 hours post dTSI (Dox, TNF, SMAC mimetic, IDN-6556; dTSI) stimulation. Red boxes denote areas of dead cells that are no longer within the discernable plane of the IncuCyte imager technology. **(B)** *MLKL*^{-/-} HT29 cells expressing MLKL^{S132P} were induced with doxycycline (Dox; 100 ng/ml) and treated with necroptotic stimulus (TNF, Smac mimetic, IDN-6556; TSI) in the presence of MLKL inhibitor NSA (1 or 10 μM). Following 8 hours of stimulation, NSA (1 or 10 μM) was administered to the cell media. Cell death was measured every hour for 24 hours by percentage of SYTOX-green positive cells using IncuCyte SX5 live cell imaging. Independent cell lines were assayed in n=4 experiments, with error bars indicating the mean ± SEM.

5. Why does the MLKL^{S132P} mutation enhances necroptosis in human cells but suppresses necroptosis in mouse cells? This is a key point that needs to be addressed. A key difference between the two systems is that the authors over-express MLKL in human cells, while the murine setting is based on a knock-in system. Hence, they are not really comparable. More data is needed here to corroborate that the human system behaves differently (e.g. more cell types, primary cells instead of cancer cells).

This is a very common and important question about this work, and one we have strived to address via multiple routes. We have now added in lines 363-367 to address the different systems as a limitation of our study. Longstanding attempts at CRISPR-mediated genomic modification to the human *MLKL* gene locus of commonly used cell lines like HT29 and U937 have not been successful to date due to very low editing efficiency and polyploidy. Primary PBMCs derived from heterozygous patients carrying the polymorphism are in very short supply thus sufficient only for the small scale, low 'n' analyses shown. EBV immortalized plasmablasts from these patients are available in abundance, but are not sensitive to necroptotic stimuli. We do not yet have access to samples from a human that is homozygous for *MLKL*^{S132P}. In the absence of genetically modified human cell lines encoding *MLKL*^{S132P}, we are limited to the following findings:

-Primary PBMCs from one heterozygous patient shows increased TNF secreted after LPS or PolyIC stimulation ex vivo, indicating a 'gain of function' (**Supplementary Figure 1B**)

-Primary PBMCs from this patient have less MLKL detectable by western blot (**Supplementary Figure 1C**). This replicates the situation for primary cells isolated from *Mlkl*^{S131P} mice.

Based on our previous work on the constitutively active mouse mutant *Mlkl*^{D139V} (Hildebrand *et al.* 2020 *Nat. Comm.*) and MLKL ubiquitination and turnover (Liu *et al.* 2021 *EMBO J*), we know that activated forms of MLKL are cleared from the cell, below a certain threshold, by a process dependent on ubiquitination. Mice that are heterozygous or homozygous for the *Mlkl*^{D139V} mutation show reduced steady state abundance of MLKL, a reduced sensitivity to necroptotic stimuli, and enhanced sensitivity to IFN- β treatment (owing to increased *Mlkl* gene expression and the accumulation of constitutively active MLKL^{D139V} mutant, surpassing the cells' clearance threshold).

6. Fig 2G suggest that IFN β /TNF treatment induces RIPK1-independent cell death in MLKL^{S132P} cells. How do these cells die? What is the involvement of RIPK3?

In mice and humans, *Mlkl*/*MLKL* is a highly interferon inducible gene. Given MLKL^{S131P} is constitutively active when expressed exogenously in mouse cells (Hildebrand *et al.* 2020 *Nat. Comm.*), endogenously encoded MLKL^{S131P} is also likely to be constitutively active. Under conditions of low basal expression, mouse cells are able to clear activated, membrane associated MLKL in a ubiquitin-dependent manner (Liu *et al.* 2021 *EMBO J*). Under conditions of high, interferon-induced expression, a cell's capacity to clear activated MLKL is surpassed, leading to cell death. We have added data to **Figure 2G** that demonstrate that IFN- β + TNF induced death in cells derived from *Mlkl*^{S131P} homozygotes is not blocked by RIPK3 inhibitor GSK'872, further supporting that this death is due to enhanced expression of constitutively active MLKL^{S131P} and not enhanced signaling via RIPK3.

7. The Incucyte data shown by the authors in the mouse cells seem to represent a quantification of PI+ cells per area. This is not appropriate to compare the different genotypes used in their experiment as it does not correct for different cell numbers in their wells. The authors should analyse this data as percentage of total cells in the well, like they do in the panels with human cells. Furthermore, across all Incucyte data, the authors should include statistics on their plots to validate the differences observed.

We recognize the reviewers concern with the presentation of PI⁺ cell quantification per mm², rather than % cell death. Calculation of % cell death via IncuCyte live cell imaging requires the concurrent use of a membrane impermeant dye (e.g. PI or SYTOX green) and a nuclear stain that stains both live and dead cells. The DRAQ5 nuclear dye used predominantly in Figure 1 and Supplementary figure 1 is toxic in mouse cells, specifically causing death after ~20 hours. Another spectrally compatible dye - SPY620 - exhibits slow uptake by murine cells and rapid dissipation of signal. In **Response Figure 5** we present data to highlight comparable plating density of all murine cell lines across genotypes. In **Response Figure 5A**, the confluences of cells plated in untreated wells at time 0 are quantified using IncuCyte analysis software. Further, in **Response Figure 5B** we present representative images of the cells plated in untreated wells at time 0 for immortalized MDF, primary MDF and bone marrow derived macrophages (BMDMs).

Response Figure 5: Equivalent plating densities for murine cells lines across genotypes.

(A) Percentage confluency of plated cells at time 0 in untreated wells. Independent cell lines were assayed in $n=2-6$ experiments. **(B)** Representative images of murine cell line plating densities at time 0 in untreated wells, with confluence mask displayed in yellow.

We thank the reviewers for their constructive comments and experimental suggestions. We are confident that we have not only thoroughly addressed their concerns in revision, but also that their suggestions have greatly improved our paper.

REVIEWER COMMENTS

Reviewer #1 (Remarks to the Author):

This is an interesting and compelling study that describes and explains the biology of common MLKL mutations. The studies are informative to the field and have been conducted with rigor. Overall, the questions and concerns that were raised during the first round of review were addressed thoroughly, either through discussion or additional experimentation. The authors should be commended on the excellent work and the fascinating study.

Reviewer #2 (Remarks to the Author):

While this rebuttal has fostered improvements to the previous manuscript, there are still several important questions that the authors have failed to appropriately address herein.

- What are the causes of the altered immune profile at steady state, in MklS131P mice?
- What causes the phenotypic differences between human and mouse MklS131P? (Only needs to be answered, if the authors wish to retain the human data in the manuscript).
- Authors show enhanced sensitivity of MklS131P cells to TNF/IFN β . The authors already show that this death occurs independent of RIPK1/RIPK3. We have no further mechanistic insight into how this death occurs. Why is TNF added if mere increase of MLKL expression should suffice?

Please see below my response to their comments/additional experiments:

1. According to my understanding, supplementary data should be considered as important as data from main figures. The link of the mutant to human disease remains poor (also see my response to point 5).

2. While the authors have attempted to offer greater phenotypic insight into the immune compartment of MklS131P/S131P, these data do not actually answer our question: 'What is the explanation for the immune alterations seen in MLKLS132P animals at steady state?' Further experimentation and/or explanations are required to resolve this point.

b. We cannot locate the ROS and Annexin V data that the authors are referring to. While interesting, the authors suggestions here remain unsubstantiated. Perhaps an inducible system would be useful in Figure 2I, to test cell death outcomes +/- IFN β treatment. Positive controls of Nec1s and RIPK3i would be needed here also, to show that these compounds are working. These are currently absent.

c. We agree that the authors have offered a fair response to this point.

3. We agree that the authors have responded appropriately to this point.

4. We agree that the authors have responded appropriately to this point.

5. While we appreciate this response, it is largely speculative. The authors need to prove that low MLKL levels are due to high turnover of MLKL. They also need to show that IFN β sensitivity to MLKLS131P cells is indeed due to MLKL – and not due to a background impact of IFN β . Also, the authors need to further interrogate the difference between human and mouse or remove the human element, as they currently cannot substantiate their claims of MLKLS131P in human cells. While the knock-in doesn't work in human, they can try overexpression in mouse systems, to see if this phenocopies human.

6. While these data do tell us that IFN β + TNF kill MklS131P homozygotes independent of RIPK3, we still do not know how they die. If the mechanism is solely MLKL-driven, then we require further readouts of MLKL activation - MLKL ubiquitylation, oligomerisation, plasma membrane localisation & cell surface membrane permeability. Without this information, we cannot definitively conclude how the cells are dying.

7. While we appreciate the explanations the authors given here, they need to find alternative means of normalising for differences in proliferation. Perhaps using a different assay (FACS) may be the way to circumvent this problem.

We are pleased that reviewer #1 is satisfied with our revised manuscript and has commended us on a thorough study. We are grateful to reviewer #2 for their time and attention and describe here a series of additional experiments, amendments and explanations that address their concerns. We have added new data in **Figure 2L and Supplementary Figure 2U, X**. We have also edited the main text in response to reviewer #2 comments and these changes are highlighted in yellow. We include below the comments of reviewer #2 *verbatim* in *blue italics*, and our responses to these in plain black text.

Response to reviewer # 2

Reviewer #2 (Remarks to the Author):

While this rebuttal has fostered improvements to the previous manuscript, there are still several important questions that the authors have failed to appropriately address herein.

- What are the causes of the altered immune profile at steady state, in MklS131P mice?*
- What causes the phenotypic differences between human and mouse MklS131P? (Only needs to be answered, if the authors wish to retain the human data in the manuscript).*
- Authors show enhanced sensitivity of MklS131P cells to TNF/IFN β . The authors already show that this death occurs independent of RIPK1/RIPK3. We have no further mechanistic insight into how this death occurs. Why is TNF added if mere increase of MLKL expression should suffice?*

Please see below my response to their comments/additional experiments:

1. According to my understanding, supplementary data should be considered as important as data from main figures. The link of the mutant to human disease remains poor (also see my response to point 5).

We completely agree with the reviewer that supplementary and main figure data are both important. Whilst we agree that the link of the S132P polymorphism and human inflammatory disease remains limited, proving unequivocal, statistically robust causation for such a high frequency polymorphism using human patients remains a significant challenge in the field of human genetics. We have embarked on a large study of thousands of patients within the UK Biobank with matched genomic and clinical data, the scale of which is beyond the scope of this manuscript but is earmarked for a follow-up study. We are confident that our wording around the caveats of having limited patient numbers (lines 372-375) are clear to readers.

2. While the authors have attempted to offer greater phenotypic insight into the immune compartment of MklS131P/S131P, these data do not actually answer our question: 'What is the explanation for the immune alterations seen in MLKLS132P animals at steady state?' Further experimentation and/or explanations are required to resolve this point.

We too share the same desire to understand the immune alterations observed in the Mkl^{S131P} mouse at steady state. We have explored many independent experimental avenues to identify the cause of these phenotypic differences in the immune cell compartment of homozygous mice;

- Multiplex assays to assess plasma cytokine profiles at steady state.
- Assessing the number/frequency of stem cell progenitors in the bone marrow.
- HSC colony forming assays.

These data have informed a hypothesis that is addressed in lines 383-390 of the discussion. Furthermore, this is an experimental question we will continue to explore but is beyond the scope of this already extensive current manuscript.

b. We cannot locate the ROS and Annexin V data that the authors are referring to.

The ROS and AnnV data were correctly located in **Supplementary Figure 3, panels C & D** in our earlier revised submission, and we have provided a screen shot below for ease of reference. These data demonstrate that in inflammatory scenarios characterized by high levels of cytokines (e.g. bone marrow post-irradiation), that are known to induce MLKL protein production, $Mkl^{S131P/S131P}$ HSCs exhibit increased signs of oxidative stress and plasma membrane disruption relative to $Mkl^{WT/WT}$ littermates. This reduces their capacity to compete and repopulate this niche. This is directly supported by our MDF data where an increased propensity for cell death is observed in homozygote cells under $IFN\beta$ treatment, that results from $MLKL^{S131P}$ constitutive activity that is independent of RIPK3.

While interesting, the authors suggestions here remain unsubstantiated. Perhaps an inducible system would be useful in Figure 2I, to test cell death outcomes +/- $IFN\beta$ treatment.

In response to the reviewer's suggestion of an exogenous system in mouse cells, we direct the reviewer to our previously published data in Hildebrand et al., 2020 *Nat. Comms*. In Figure 5 panels c and d (displayed below), we previously showed that in an exogenous expression system simply inducing expression of $MLKL^{S131P}$ is sufficient for formation of high molecular weight membrane associated complexes and resultant death of cells. This published observation is consistent with the constitutive activity we observe in the endogenous system (**Figure 2L**),

where we posit that IFN β addition mimics the role of doxycycline in the exogenous system by triggering *Mikl* gene expression above a threshold.

Positive controls of Nec1s and RIPK3i would be needed here also, to show that these compounds are working. These are currently absent.

We thank the reviewer for raising this oversight. We have now included relevant control samples (TSI + GSK-872 or Nec-1s) in **Figure 2L**, which show significant reductions in TSI induced cell death but show no alleviation of IFN β induced cell death. These data also demonstrate that a greater portion of the death induced by TSI in *Mikl*^{WT/S131P} and *Mikl*^{S131P/S131P} cells is refractory to GSK-872 and Nec-1-s, reflecting TNF-mediated increases in *Mikl* gene expression and constitutive activity akin to that observed with IFN β . A screen shot of this data is placed below for reviewer convenience.

Figure 2

- c. We agree that the authors have offered a fair response to this point.*
- 3. We agree that the authors have responded appropriately to this point.*
- 4. We agree that the authors have responded appropriately to this point.*

We thank the reviewer for their appraisal of points 2c, 3 and 4.

5. While we appreciate this response, it is largely speculative. The authors need to prove that low MLKL levels are due to high turnover of MLKL. They also need to show that IFN β sensitivity to of MLKLS131P cells is indeed due to MLKL – and not due to a background impact of IFN β .

We thank the reviewer for prompting further clarification of these important points and below we have provided a response to each question raised.

We strongly believe that side-by-side examination of WT/WT cells alongside S131P/S131P cells isolated from littermate controls in these experiments (where the only experimental variable is the genotype of MLKL), sufficiently demonstrates that the death observed is indeed MLKL-dependent. However, to further exclude an MLKL-independent background effect, below we provide cell death data from *Mlkl*^{-/-} MDFs stimulated with IFN β in the presence or absence of TNF. We observed no cell death under IFN β treatment alone or in conjunction with TNF in MDFs with genetic knockout of *Mlkl*. These data are presented below in **Response Figure 1**.

Furthermore, in our response to point 6 (below) we also show the formation of high molecular weight MLKL complexes in the crude membrane fraction of *Mlkl*^{S131P/S131P} MDFs when stimulated with IFN β and TNF alone, supporting a more direct role of MLKL under these conditions.

Response Figure 1. *Mlkl*^{-/-} MDFs were stimulated as denoted and cell death was measured as a count of SYTOX Green positive cells (per mm²) every hour for 24 hours using IncuCyte SX5 imaging technology. Under apoptotic stimulation (TNF, Smac mimetic; TS) *Mlkl*^{-/-} MDFs undergo cell death indicated by an increase in the count of SYTOX Green positive cells. Following necroptotic stimulation (TNF, Smac mimetic, IDN-6556; TSI), IFN β , or IFN β + TNF treatment no observable cell death was measured in *Mlkl*^{-/-} MDFs. Data shown mean \pm SEM of $n = 3$ independently generated cell lines.

Also, the authors need to further interrogate the difference between human and mouse or remove the human element, as they currently cannot substantiate their claims of MLKLS131P in human cells. While the knock-in doesn't work in human, they can try overexpression in mouse systems, to see if this phenocopies human.

As outlined above in **point 2b**, we have indeed tested the exogenous expression of *Mlkl*^{S131P} in mouse cells and find that exogenous expression of *Mlkl*^{S131P} in *Mlkl*^{-/-} MDFs results in constitutive cell death. Therefore, in both exogenous and endogenous systems *Mlkl*^{S131P} exhibits constitutive activity. Expression of human MLKL^{S132P} and mouse *Mlkl*^{S131P} in respective cell lines both result in **gain-of-function** and **enhance** cell death. In human cells this manifests in a gain-of-function under specific inhibitory conditions (NSA or Serine83 mutation) and in mouse cells (both endogenous and exogenous expression) MLKL^{S131P} exhibits constitutive activity. These nuanced differences in the manifestation of the gain-of-function are consistent with the

current consensus in the field that mouse and human MLKL have underlying differences in their activation and execution of necroptosis¹⁻⁶. The “kiss and run” mechanism of mouse Mlkl activation lends itself to a greater capacity for constitutive activity, as its requirement for RIPK3 interaction is transient and fulfilled following phosphorylation^{2, 7-9}. In contrast, it is well established that human MLKL in contrast requires a stable interaction with RIPK3 to execute necroptosis^{4, 5}.

6. While these data do tell us that IFN β + TNF kill *Mlkl*^{S131P} homozygotes independent of RIPK3, we still do not know how they die. If the mechanism is solely MLKL-driven, then we require further readouts of MLKL activation - MLKL ubiquitylation, oligomerisation, plasma membrane localisation & cell surface membrane permeability. Without this information, we cannot definitively conclude how the cells are dying.

We thank the reviewer for their suggestion and agree it is important to demonstrate the activity of MLKL^{S131P} following IFN β + TNF stimulation more directly. As mentioned earlier, we see a significant shift in the ratio of low molecular weight to high molecular weight MLKL complexes in the crude membrane fraction of *Mlkl*^{S131P/S131P} MDFs when stimulated with IFN β and TNF when compared to *Mlkl*^{WT/WT} MDFs. This high molecular weight complex resembles that formed when cells are stimulated with TSI. These data have now been added to **Supplementary Figure 2X** and are displayed below for reviewer convenience. We highlight that this is consistent with the BN-PAGE data presented in Hildebrand et al., 2020 *Nat. Comms*. Figure 5c, where exogenous expression of Mlkl^{S131P} spontaneously forms membrane-associated high molecular weight complexes (displayed above in point 2b).

We also note that the precise role of MLKL ubiquitylation in its activation mechanism is currently an active area of investigation, with conflicting roles assigned to activation and suppression^{10, 11}. Because MLKL ubiquitylation as a marker for activation is disputed, perhaps because different sites serve different functions in distinct contexts, we have elected not to examine ubiquitylation as a readout for activation here.

7. While we appreciate the explanations the authors given here, they need to find alternative means of normalising for differences in proliferation. Perhaps using a different assay (FACS) may be the way to circumvent this problem.

We are very confident that differences in proliferative capacity do not explain the differences observed here over a 24-hour period, where we can demonstrate comparable proliferation across time. Below we show percentage confluence measurements across 0-24 hours for untreated immortalized MDFs (**Response Figure 2**) and clearly no significant differences are observed in the proliferative capacity of MDFs of different genotypes. We have made a conscious transition to real-time imaging of adherent cells in place of FACS analysis because FACS offers only a snapshot at a single timepoint, while IncuCyte allows temporal analysis. Additionally, IncuCyte is more robust because it obviates the need for additional stress to membrane integrity that is introduced by the process of enzymatic detachment of adherent cells (required for FACS).

Response Figure 2. Confluence measurements from 0-24 hours for untreated immortalized $Mkl^{WT/WT}$, $Mkl^{WT/S131P}$ and $Mkl^{S131P/S131P}$ MDFs. Percentage confluence was measured using IncuCyte SX5 imaging technology. Data shown mean \pm SEM of $n = 6-10$ independent repeats.

1. Davies KA, Fitzgibbon C, Young SN, Garnish SE, Yeung W, Coursier D, Birkinshaw RW, Sandow JJ, Lehmann WIL, Liang LY *et al.* (2020) Distinct pseudokinase domain conformations underlie divergent activation mechanisms among vertebrate MLKL orthologues. *Nat Commun.* 11(1):3060.
2. Davies KA, Tanzer MC, Griffin MDW, Mok YF, Young SN, Qin R, Petrie EJ, Czabotar PE, Silke J, Murphy JM. (2018) The brace helices of MLKL mediate interdomain communication and oligomerisation to regulate cell death by necroptosis. *Cell Death Differ.*
3. Murphy JM. (2020) The Killer Pseudokinase Mixed Lineage Kinase Domain-Like Protein (MLKL). *Cold Spring Harb Perspect Biol.* 12(8).
4. Petrie EJ, Sandow JJ, Jacobsen AV, Smith BJ, Griffin MDW, Lucet IS, Dai W, Young SN, Tanzer MC, Wardak A *et al.* (2018) Conformational switching of the pseudokinase domain promotes human MLKL tetramerization and cell death by necroptosis. *Nat Commun.* 9(1):2422.
5. Tanzer MC, Matti I, Hildebrand JM, Young SN, Wardak A, Tripaydonis A, Petrie EJ, Mildenhall AL, Vaux DL, Vince JE *et al.* (2016) Evolutionary divergence of the necroptosis effector MLKL. *Cell Death Differ.* 23(7):1185-1197.
6. Petrie EJ, Czabotar PE, Murphy JM. (2019) The Structural Basis of Necroptotic Cell Death Signaling. *Trends Biochem Sci.* 44(1):53-63.
7. Murphy JM, Czabotar PE, Hildebrand JM, Lucet IS, Zhang JG, Alvarez-Diaz S, Lewis R, Lalaoui N, Metcalf D, Webb AI *et al.* (2013) The pseudokinase MLKL mediates necroptosis via a molecular switch mechanism. *Immunity.* 39(3):443-453.
8. Tanzer MC, Tripaydonis A, Webb AI, Young SN, Varghese LN, Hall C, Alexander WS, Hildebrand JM, Silke J, Murphy JM. (2015) Necroptosis signalling is tuned by phosphorylation of MLKL residues outside the pseudokinase domain activation loop. *Biochem J.* 471(2):255-265.
9. Hildebrand JM, Tanzer MC, Lucet IS, Young SN, Spall SK, Sharma P, Pierotti C, Garnier JM, Dobson RC, Webb AI *et al.* (2014) Activation of the pseudokinase MLKL unleashes the four-helix bundle domain to induce membrane localization and necroptotic cell death. *Proc Natl Acad Sci U S A.* 111(42):15072-15077.
10. Liu Z, Dagley LF, Shield-Artin K, Young SN, Bankovacki A, Wang X, Tang M, Howitt J, Stafford CA, Nachbur U *et al.* (2021) Oligomerization-driven MLKL ubiquitylation antagonizes necroptosis. *EMBO J.* 40(23):e103718.
11. Garcia LR, Tenev T, Newman R, Haich RO, Liccardi G, John SW, Annibaldi A, Yu L, Pardo M, Young SN *et al.* (2021) Ubiquitylation of MLKL at lysine 219 positively regulates necroptosis-induced tissue injury and pathogen clearance. *Nat Commun.* 12(1):3364.

REVIEWERS' COMMENTS

Reviewer #2 (Remarks to the Author):

The authors have made a genuine effort to address the remaining issues and have addressed these either experimentally or through text changes. I am happy with the revisions and respective text changes.